

# Entanglement in the quantum Hall fluid of dipoles

**Jackson R. Fliss**⋆

Institute of Physics, University of Amsterdam,
904 Science Park, 1098 XH Amsterdam, The Netherlands

⋆ j.r.fliss@uva.nl

## Abstract

We revisit a model for gapped fractonic order in (2+1) dimensions (a symmetric-traceless tensor gauge theory with conservation of dipole and trace-quadrupole moments described in [1]) and compute its ground-state entanglement entropy on $\mathbb{R}^2$. Along the way, we quantize the theory on open subsets of $\mathbb{R}^2$ which gives rise to gapless edge excitations that are Lifshitz-type scalar theories. We additionally explore varieties of gauge-invariant extended operators and rephrase the fractonic physics in terms of the local deformability of these operators. We explore similarities of this model to the effective field theories describing quantum Hall fluids: in particular, quantization of dipole moments through a novel compact symmetry leads us to interpret the vacuum of this theory as a dipole condensate atop of which dipoles with fractionalized moments appear as quasi-particle excitations with Abelian anyonic statistics. This interpretation is reflected in the subleading "topological entanglement" correction to the entanglement entropy. We extend this result to a series of models with conserved multipole moments.

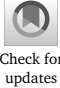

# 1  Introduction

The history of using quantum field theory as a tool for describing low energy phases of matter is one with many successes. Much of the confidence in this program relies on the "low-energy dogma" that at broad scales and low-energies the essential physics forgets microscopic details such as the lattice spacing. This dogma has been formalized through the renormalization group: as characteristic energy scales are lowered, the irrelevant couplings of a putative microscopic Hamiltonian run to weak coupling. The IR fixed point of this flow is a low-energy effective field theory whose Lagrangian consists of all possible relevant interactions allowed by symmetry and which characterizes a universality class of microscopic theories.

One of the most notable successes of this dogma is the characterization of $(2+1)$-d gapped phases of matter with topological order. The low-energy descriptions of such phases are topological quantum field theories which have been wildly successful at characterizing their universal signatures: gapless edge states [2], topologically robust ground-state degeneracy [3,4], anyonic quasi-particle statistics [5], long-range entanglement of their ground states [6,7], and more modernly, the spontaneous breaking of a higher-form symmetries [8]. Given this, it is a reasonable expectation that topological field theories provide a unifying low-energy description of all gapped phases of matter in any dimension.

Contrary to this expectation was the discovery of $(3+1)$-d Hamiltonians [9–13] with features that are at odds with the low-energy dogma. While displaying some typical features of topological order (e.g. gapped spectrum, robust ground state degeneracy), these models are distinguished by the emergence of *fractons*: excitations whose mobility is either locked in position or forced to move along subdimensional manifolds. These fractonic excitations can either be freed by the introduction of additional excitations or they can be completely locked. Other common features include an extensive degeneracy of ground states (even at zero temperature) and symmetries associated to subdimensional manifolds. Thus despite arising from rather tame, local, stabilizer Hamiltonians, these features of fractonic phases make them difficult to encompass in the framework of quantum field theory, at least as typically presented.

Recently significant progress has been made in writing down and studying "fractonic field theories" by either relaxing basic notions of quantum field theory (such as allowing states with divergent energy in the continuum limit [14–16]) or supplementing them with extra geometric structure (such as a background foliation [17,18] or constraints inherited from a lattice description [19]). Complementary to these approaches are a set of *gapless* models pioneered by Pretko [20,21] described by fairly standard continuum field theories, while also displaying key characteristics of fractonic physics. These models are collectively based on *tensor gauge theories* that explicitly break Lorentz symmetry. The spatial tensor structure of the gauge constraints enforces multi-polar conservation laws that provide a natural mechanism for locking charge excitations in place.

To be specific, let us introduce the "scalar-charge theory" in $D$ spatial dimensions, which

has a symmetric two-tensor "electric field" obeying a Gauss-law constraint

$$\partial_i \partial_j E^{ij} = \rho \,. \tag{1}$$

Introducing a scalar and symmetric tensor potential $\{A_0, A_{ij}\}$ through $E_{ij} = \partial_t A_{ij} - \partial_i \partial_j A_0$, we can construct a corresponding "magnetic field", $B^{i_1 i_2 \dots i_{D-2}}{}_j = \varepsilon^{i_1 i_2 \dots i_D} \partial_{i_{D-1}} A_{i_D j}$ and the dynamics of this theory is governed by a "tensor Maxwell" action

$$S_{tM} = \frac{1}{2g^2} \int dt \int d^D x \left( E^{ij} E_{ij} - B^{i_1 i_2 \dots i_{D-2} j} B_{i_1 i_2 \dots i_{D-2} j} \right) + \int dt \int d^D x \left( A_0 \rho + A_{ij} J^{ij} \right). \tag{2}$$

The cost of introducing $A_0$ and $A_{ij}$ is a gauge-redundancy of the form

$$A_0 \to A_0 + \partial_t \alpha \,, \qquad\qquad A_{ij} \to A_{ij} + \partial_i \partial_j \alpha \,, \tag{3}$$

which leads to the conservation law

$$\partial_t \rho - \partial_i \partial_j J^{ij} = 0 \,. \tag{4}$$

As a consequence, in addition to the charge, the integrated dipole moment on the plane is conserved due to the additional integration by parts:

$$\partial_t Q_0 = \partial_t \int d^D x \, \rho = 0 \,, \qquad\qquad \partial_t Q_1^i = \partial_t \int d^D x \, (x - x_0)^i \rho = 0 \,. \tag{5}$$

This additional conservation of dipole moment is the meat of the fractonic physics of this tensor gauge theory: individual charges are locked in place because their motion would violate dipole conservation. Two charges of opposite sign, however, regain mobility as long as they move in tandem as a dipole pair.

While (5) is true in any dimension, let us draw some connections between the scalar charge tensor gauge theory in $D = 2$ dimensions and other well-known lattice models of dipole-conserving fractonic order. The first is referred to as the *XY-plaquette model* [22] on a square lattice, $\mathfrak{L}$, with canonical variables $\pi$ and $\phi$ and Hamiltonian

$$H_{XY} = \sum_{\hat{r} \in \mathfrak{L}} \left( \pi_{\hat{r}} \pi_{\hat{r}} - K \cos(\hat{\Delta}_{xy} \phi_{\hat{r}}) \right), \tag{6}$$

where $\Delta_{ij} \phi_{\hat{r}} = \phi_{\hat{r} + \hat{e}_i + \hat{e}_j} - \phi_{\hat{r} + \hat{e}_i} - \phi_{\hat{r} + \hat{e}_j} + \phi_{\hat{r}}$ and $\{\hat{e}_i\} = \{\hat{e}_x, \hat{e}_y\}$ generate the lattice. This model has an extensive number of $U(1)$ global symmetries ($L_x + L_y - 1$ for an $L_x \times L_y$ lattice with periodic boundary conditions) however these symmetries are explicitly broken by adding the additional interaction:

$$\delta H = -\frac{K'}{2} \sum_{\hat{r} \in \mathfrak{L}} \left( \cos(\hat{\Delta}_{xx} \phi_{\hat{r}}) + \cos(\hat{\Delta}_{yy} \phi_{\hat{r}}) \right), \tag{7}$$

leaving only a global shift symmetry and a global dipolar shift symmetry [23]. These global symmetries can be gauged by coupling their currents to lattice gauge fields with tensor structure, which is described by a tensor gauge theory in the continuum [24]. Note that the $K' = K$ point leads to a rotationally symmetric model in the continuum and lowest order action respecting the global symmetries and this rotational symmetry is (2).

Secondly, the 2D scalar-charge theory has connections to the quantum theory of elasticity in two-dimensional lattices [25, 26]. In this dictionary the tensor electric field $E_{ij}$ maps to the dual of the symmetric strain tensor, $\varepsilon_i{}^k \varepsilon_j{}^l u_{kl}$, and the magnetic field to the dual of its conjugate momentum. The gapless excitations of the tensor gauge theory describe the transverse

and longitudinal phonons of the lattice. An important entry into this dictionary is the identification of fracton charges with *disclinations* which cannot move without introducing additional excitations. Bound pairs of disclinations form *dislocations* which map to the mobile dipoles of the tensor-gauge theory. In this identification, the dipole moment is orthogonal to the Burgers vector of the dislocation:

$$d^i = \varepsilon^i{}_j b^j. \tag{8}$$

The dipole can "glide" along this vector (that is, orthogonal to its dipole moment), but motion perpendicular to $\vec{b}$, "the climb," requires the presence of energetically costly vacancies. We might then expect the low-energy limit of this model to have restricted dipole motion in addition to fracton charges. We will see this shortly.

In this paper we explore a *gapped* model of fractonic order derived from $(2+1)$-d tensor gauge theory by adding a Chern-Simons-like term [1] [1]. In the cited paper, the authors noted that the subsequent gapped theory displays both fractonic excitations (associated to the conservation of dipole moment discussed above) as well characteristics familiar to the Abelian Chern-Simons descriptions of quantum Hall fluids, leading the authors to coin it a *"dipolar quantum Hall fluid."* More recently, the connection between tensor gauge theories and quantum Hall physics has been noticed in [28]. The central aim of the present paper is to make this connection more explicit and in a language familiar to high-energy physicists. In doing so we will explicate several features of this model.

In section 2 we will introduce the action and its associated gauge symmetries; we go on to quantize the theory on $\mathbb{R}^2$ in section 2.1 and write down its vacuum wave-function. In section 3 we investigate the types of gauge-invariant operators in the theory which are, by nature, extended. In addition to the somewhat typical 1-dimensional line operators, we also discuss two classes of gauge-invariant operators that are allowed to be extended (either fully or finitely) in a second dimension which we collectively call *strip operators*. The strip operators provide an alternative characterization of the fractonic features of this model through constraints on their local deformability.

In [1], the authors take the charges associated to dipole moments to be quantized. We provide an interpretation of this dipole quantization in section 4 in terms of an invariance of the theory under a set of "large gauge transformations" that are elements of a compact symmetry group associated to an underlying lattice. This dovetails with a similar occurrences of lattice-organized symmetries in other fracton models and we provide coarse justifications for including this symmetry in section 4. The subsequent charge quantizations have drastic physical implications for the vacuum of the theory beyond that of a simple insulating phase. In particular we find that the vacuum forms a condensate that allows "long" dipoles to become "transparent" and fall into the condensate. As we explain in section 4.1, this condensate restores mobility to "short" dipoles, which we regard as the fundamental quasi-particle excitations, and which obtain anyonic statistics.

Lastly, we perform a calculation of the entanglement entropy of the ground-state of this dipolar condensate and show that it takes the form of two separate Abelian topological orders (associated to the two independent dipole orientations). We regard this as the major result of this paper: to our knowledge this is both the first calculation of the entanglement entropy of a tensor gauge theory as well as the first entanglement entropy calculation of a fractonic model using continuum quantum field theory techniques. We additionally recast this calculation from the perspective of edge modes (section 5.2) and propose an expectation for the ground state entanglement of a series of gapped $(2+1)$-d fracton models with conserved multipole moments. We briefly discuss the physics of these results and their implications for future research in section 6. Lastly, in the appendix, we provide arguments for the universality of our

---

[1]Historically, this term was first introduced as a boundary action in [27].

answer for the entanglement entropy (appendix A), and details on large gauge transformations (appendix B).

## 2  Tensor Chern-Simons theory

Let us now introduce the "tensor Chern-Simons" term in $(2+1)$-dimensions from [1]

$$S_{tCS} = \frac{k}{2\pi} \int dt \int_\Sigma d^2x\, A_0\, \varepsilon^{ij}\delta^{kl}\partial_i\partial_k A_{jl} + \frac{k}{4\pi}\int dt\int_\Sigma d^2x\, \varepsilon^{ij}\delta^{kl} A_{ik}\partial_t A_{jl}. \qquad (9)$$

This model is not topological in the conventional sense[2] although we will see that it shares many characteristics of a traditional topological gapped phase. It is simple to check that the action only involves the symmetric trace-less part of $A_{ij}$:

$$A_{ij}^{S.T.L.} = A_{(ij)} - \frac{1}{2}\delta_{ij}\delta^{kl}A_{kl}, \qquad\qquad S_{tCS}[A] = S_{tCS}[A^{S.T.L.}]. \qquad (11)$$

As explained in [1], the trace of $A_{ij}$ can be removed by a suitable gauge transformation affecting only the tensor-Maxwell term. Thus the degrees of freedom for the trace of $A_{ij}$ are described by the action $S_{tM}$ whose coupling is irrelevant in $(2+1)$ dimensions. Taking an IR perspective, and we will path-integrate only the symmetric-traceless configurations with action (9) regarding them as the relevant degrees of freedom:

$$Z = \int \frac{\mathcal{D}A_0\,\mathcal{D}A_{ij}^{S.T.L}}{\mathcal{V}_g} e^{iS_{tCS}[A_0, A_{ij}^{S.T.L}]}. \qquad (12)$$

Here onward we will drop the superscript "$S.T.L.$" with symmetric-traceless being understood.

We indicate schematically by $\frac{\mathcal{D}A_0\mathcal{D}A_{ij}}{\mathcal{V}_g}$ that the measure should be considered as an integration over orbits of the Abelian gauge symmetry which has been modified due to the elimination of the trace from the field variable:

$$A_{ij} \to A_{ij} + \left(\partial_i\partial_j - \frac{1}{2}\delta_{ij}\partial^2\right)\alpha, \qquad\qquad A_0 \to A_0 + \dot{\alpha}. \qquad (13)$$

Corresponding to this gauge symmetry are conserved charges found by coupling to a charge-density $\rho$ and a symmetric-traceless current $J^{ij}$

$$S[\rho, J^{ij}] = S_{tCS} + \int dt \int_\Sigma d^2x\, A_0\,\rho + \int dt \int_\Sigma d^2x\, A_{ij}J^{ij}, \qquad (14)$$

which must obey the following conservation law to maintain gauge invariance under (13):

$$\dot{\rho} - \left(\partial_i\partial_j - \frac{1}{2}\delta_{ij}\partial^2\right)J^{ij} = 0. \qquad (15)$$

---

[2]Naïvely one might think that this theory couples naturally to a background metric, $g$, on $\Sigma$:

$$S_{tCS}[g,A] = \frac{k}{2\pi}\int dt \int_\Sigma d^2x\,\sqrt{g}\,A_0\,\varepsilon^{ij}g^{kl}\nabla_i\nabla_k A_{jl} + \frac{k}{4\pi}\int dt\int_\Sigma d^2x\,\sqrt{g}\,\varepsilon^{ij}g^{kl}A_{ik}\partial_t A_{jl}, \qquad (10)$$

which makes its geometric dependence evident, however as emphasized by Gromov [26] such a theory is inconsistent even with weak curvature (see [29] for an alternative perspective). Although it can be embedded into a sensible theory that couples to geometric curvature, for the purposes of this paper we will stick to $\Sigma = \mathbb{R}^2$ or a subset of $\mathbb{R}^2$ with the Euclidean metric.

As a local conservation law (15) is valid on all backgrounds and is the essence of the fractonic character of this model[3] however it is useful to state this in terms of (background dependent) global quantities. Namely, there is conservation of global charge:

$$\partial_t Q_0 = \partial_t \left( \int_\Sigma d^2 x \rho \right) = 0 \,, \tag{16}$$

which is conserved on any $\Sigma$ without boundary. When $\Sigma = \mathbb{R}^2$, we have two additional global conserved quantities: the dipole moment,

$$\partial_t Q_1^i[\vec{x}_0] = \partial_t \left( \int d^2 x \, (x - x_0)^i \rho \right) = 0 \,, \tag{17}$$

and the *trace* of the quadrupole moment:

$$\partial_t \left( Q_2^T[\vec{x}_0] \right) = \partial_t \left( \delta_{ij} \int d^2 x \, (x-x_0)^i (x-x_0)^j \rho \right) = \delta_{ij} \int d^2 x \left( 2 J^{ij} - \delta^{ij} \delta_{kl} J^{kl} \right) = 0 \,, \tag{18}$$

which follows from the elimination of the trace from $A_{ij}$. The generic quadrupole moment is not conserved. Thus much like the tensor Maxwell theory, individual charged excitations are fractonic: they are locked in position by the conservation of the dipole moment but can be freed when forming a dipole pair. However in a departure from the tensor-Maxwell theory, dipoles are not fully mobile. They are restricted to move transverse to their dipole moment as a consequence of conserving the trace-quadrupole moment.

More generally one can imagine constructing a whole series of gapped fractonic models based upon a slight generalization of (9) to an action involving a symmetric-traceless $q$-tensor, $A_{i_1 i_2 \ldots i_q}$:

$$S_{q-tCS} = \frac{k}{4\pi} \int dt \int_\Sigma d^2 x \, \varepsilon^{i_1 j_1} \delta^{i_2 j_2} \ldots \delta^{i_q j_q} \left( 2 A_0 \partial_{i_1} \partial_{i_2} \ldots \partial_{i_q} A_{j_1 j_2 \ldots j_q} + A_{i_1 i_2 \ldots i_q} \partial_t A_{j_1 j_2 \ldots j_q} \right) , \tag{19}$$

which has a gauge invariance of

$$\delta_\alpha A_0 = \partial_t \alpha \,, \qquad \delta_\alpha A_{i_1 i_2 \ldots i_q} = \partial_{i_1} \partial_{i_2} \ldots \partial_{i_q} \alpha - \text{traces} \,, \tag{20}$$

which by a similar mechanism locks in the $(q-1)$ multipole moments. The $q^{th}$ multipole is mobile but its mobility is restricted by the tracelessness condition. This model shares many of the same moral characteristics of the dipolar $q = 2$ theory, however for the sake of definiteness we will content ourselves with a focused discussion of the $q = 2$ for the rest of this paper and only comment briefly on the $q$-tensor model at various points when appropriate. Importantly, we will propose the expected entanglement entropy of the $q$-tensor ground state in section 5.

Returning to the $q = 2$ model, all equations of motion are constraints. In particular we can perform the $\mathcal{D}A_0$ path-integration in (12) which yields a delta-functional enforcing a "tensor Gauss's law" constraint:

$$\frac{k}{2\pi} \varepsilon^{ik} \delta^{jl} \partial_i \partial_j A_{kl} = \rho \,. \tag{21}$$

The remaining equation of motion for the $A_{ij}$

$$\frac{k}{2\pi} \delta^{k(j} \varepsilon^{i)l} \partial_t A_{kl} = J^{ij} \,, \tag{22}$$

is first order and as such there are no propagating degrees of freedom: this model is completely gapped.

---

[3]In section 3 we will phrase this fractonic character as conditions on the *local* deformability of defect operators.

Lastly let us take a brief moment to comment on engineering dimensions. The fields have length dimensions

$$[A_{ij}] \sim \ell^{-1}, \qquad [A_0] \sim \ell^0, \tag{23}$$

and so the coupling $k$ is dimensionless[4]. Importantly it follows that the gauge parameters in (13) have dimension $[\alpha] \sim \ell$ and sources in (14) have dimension $[\rho] \sim \ell^{-3}$ and $[J^{ij}] \sim \ell^{-2}$. We comment that the interpretation of $\rho$ as a (spatial) charge density requires the introduction of a supplemental length scale. We will elaborate further on this below when the need arises.

## 2.1 Quantization and the vacuum wave-function

Now we focus on the preparation of states via the path-integral of the theory on $\mathbb{R}_- \times \Sigma$, where Euclidean time ranges from -∞ to 0. Given the usual interpretation in quantum field theory, this path integral will construct for us a wave-functional on $\Sigma$, taking the boundary conditions at $t = 0$ as input. Although we are interested in $\mathbb{R}^2$, we write $\Sigma$ when it is possible to keep manipulations general and then specialize to $\mathbb{R}^2$ near the end. We will also take this chance to also discuss quantization on a general surface $\Sigma$. We begin by considering variations of the action around classical configurations. This defines for us the (pre)symplectic one-form:

$$\theta[A, \delta A] := \delta S_{tCS}|_{\text{on-shell}} = \frac{k}{4\pi} \int_{\Sigma} d^2x \, \varepsilon^{ij} \delta^{kl} A_{ik} \left(\delta A_{jl}\right). \tag{24}$$

Note that the portion of the action involving $A_0$ has only spatial derivatives and $A_0$ does not appear as a canonical degree of freedom. Its role is to enforce a "tensor-Gauss law" constraint:

$$\frac{k}{2\pi} \varepsilon^{ij} \delta^{kl} \partial_i \partial_k A_{jl} = 0. \tag{25}$$

We will find it useful in this discussion to also consider the more general case of coupling $A_0$ to a collection point-source defects, $\{q^a\}$, on $\Sigma$ of the form $\rho(\vec{x}) = \mu \sum_a q^a \delta^2(\vec{x} - \vec{\mathbf{x}}_a)$. Here $\mu^{-1}$ is a length scale introduced to make $q^a$ dimensionless. We will elaboration on the interpretation of this length scale in section 4. This modifies (25) to

$$\frac{k}{2\pi} \varepsilon^{ij} \delta^{kl} \partial_i \partial_k A_{jl} = \mu \sum_a q^a \delta^2(\vec{x} - \vec{\mathbf{x}}_a). \tag{26}$$

The second variation of the action gives us the (pre)symplectic form:

$$\Omega = \frac{k}{4\pi} \int_{\Sigma} d^2x \, \varepsilon^{ij} \delta^{kl} (\delta A_{ik})(\delta A_{jl}) \tag{27}$$

the inverse of which determines the classical Poisson brackets of the theory:

$$\left\{A_{ik}(t, \vec{x}), A_{jl}(t, \vec{y})\right\}_{P.B.} = \frac{4\pi}{k} \varepsilon_{\langle ij} \delta_{kl \rangle} \delta_{\Sigma}^2(\vec{x} - \vec{y}), \tag{28}$$

where we introduce the subscript shorthand $\langle i_1 i_2 i_3 i_4 \rangle$ on the right-hand side to indicate taking the symmetric combinations under $i_1 \leftrightarrow i_3$ and $i_2 \leftrightarrow i_4$. The quantization then proceeds simply by the promotion of the fields to operators and equipping them with canonical commutators

$$\left[\hat{A}_{ik}(t, \vec{x}), \hat{A}_{jl}(t, \vec{y})\right] = i \frac{4\pi}{k} \varepsilon_{\langle ij} \delta_{kl \rangle} \delta_{\Sigma}^2(\vec{x} - \vec{y}), \tag{29}$$

or in components

$$\left[\hat{A}_{xx}(t, \vec{x}), \hat{A}_{xy}(t, \vec{y})\right] = -\left[\hat{A}_{yy}(t, \vec{x}), \hat{A}_{xy}(t, \vec{y})\right] = i \frac{2\pi}{k} \delta_{\Sigma}^2(\vec{x} - \vec{y}). \tag{30}$$

---

[4]while the tensor-Maxwell coupling has dimensions $[1/g^2] \sim \ell$ and is irrelevant.

The fields are more conveniently quantized in a holomorphic form. Indeed, returning to the pre-symplectic one-form, (24), we note it is currently in a mixed form (for instance, in standard coordinates it is roughly $\theta \sim \int A_{xx}\delta A_{xy} - A_{xy}\delta A_{xx}$ after eliminating $A_{yy}$ using the traceless condition). In order to put this into Darboux form $\theta \sim \int \mathbf{p}\,\delta\mathbf{q}$ for an appropriate $\mathbf{p}$ and $\mathbf{q}$ we need to add a suitable boundary term. This would then tell us that fixing the $t = 0$ value of $\mathbf{q}$ is consistent with the variational principle. Taking a hint from ordinary Chern-Simons theory, a suitable boundary term is given by

$$S^{\pm}_{bndy} = \pm i\frac{k}{8\pi}\int_{\Sigma} d^2x\, \delta^{kl}\left(A_{ik}dx^i\right) \wedge \star \left(A_{jl}dx^j\right), \tag{31}$$

where $\star$ is the Hodge star compatible with the flat metric on $\Sigma$. Because $\Sigma$ is Euclidean, $(i\star)$ is an involution on 1-forms (i.e. $(i\star)^2 = 1$). Thus, at least locally, we can find coordinates $\{z,\bar{z}\}$ such that $i \star dz = dz$ and $i \star d\bar{z} = -d\bar{z}$ and $(1 \pm i\star)$ acts as projector. In these local coordinates $\delta_{ij}dx^i dx^j = dz d\bar{z}$ and so tracelessness implies $A_{z\bar{z}} = 0$. The inclusion of this boundary term then shifts the pre-symplectic one-form to

$$\theta^{\pm}[A,\delta A] = \begin{cases} -\frac{k}{2\pi}\int_{\Sigma} dz d\bar{z}\, A_{\bar{z}\bar{z}}\delta A_{zz}, & + \\ +\frac{k}{2\pi}\int_{\Sigma} dz d\bar{z}\, A_{zz}\delta A_{\bar{z}\bar{z}}, & - \end{cases} \tag{32}$$

and so this is consistent with fixing either $A_{zz}$ or $A_{\bar{z}\bar{z}}$ at $t = 0$, respectively. Let us choose the first option, fixing $A_{zz}$ on $\Sigma$. The symplectic two-form is

$$\Omega = \frac{k}{2\pi}\int_{\Sigma} d^2z\, \delta A_{zz}\delta A_{\bar{z}\bar{z}}, \tag{33}$$

which leads to the commutation relations

$$[\hat{A}_{zz}(\vec{x}_1),\hat{A}_{\bar{z}\bar{z}}(\vec{x}_2)] = i\frac{2\pi}{k}\delta^2(\vec{x}_1 - \vec{x}_2). \tag{34}$$

From here let us discuss the wave-function. We will simplify the notation by $A = A_{zz}$ and $\bar{A} \equiv A_{\bar{z}\bar{z}}$ and will also denote $\partial \equiv \partial_z$ and $\bar{\partial} \equiv \partial_{\bar{z}}$. Path-integration will produce a wave-functional of the boundary value of $a$. Let us examine this wave-functional. To be general we will incorporate charge defects on $\Sigma$ of the form (26). Let us write $A_{ij} = A_{ij}^{(0)} + B_{ij}$ with $A_{ij}^{(0)}$ a fixed time-independent background configuration satisfying $\partial^2 \bar{A}^{(0)} - \bar{\partial}^2 A^{(0)} = \mu\frac{2\pi}{k}\sum_a q^a \delta^2(z - \mathbf{z}_a, \bar{z} - \bar{\mathbf{z}}_a) \equiv \frac{2\pi}{k}\rho$. One such configuration is given by

$$A^{(0)} = -\frac{\mu}{k}\sum_a q^a \frac{(\bar{z} - \bar{\mathbf{z}}_a)}{(z - \mathbf{z}_a)}, \qquad \bar{A}^{(0)} = \frac{\mu}{k}\sum_a q^a \frac{(z - \mathbf{z}_a)}{(\bar{z} - \bar{\mathbf{z}}_a)}, \tag{35}$$

which satisfies $\partial^2 \bar{A}^{(0)} = -\bar{\partial}^2 A^{(0)} = \mu\frac{\pi}{k}\sum_a q^a \delta^2(z - \mathbf{z}_a, \bar{z} - \bar{\mathbf{z}}_a)$ distributionally. Denoting the boundary value $B|_{\Sigma} = b$, the functional obtained by path-integration on $\mathbb{R}_- \times \Sigma$ is:

$$\psi[b] := \int \frac{\mathcal{D}B_{ij}}{\mathcal{V}_g}\bigg|_{B[\Sigma]=b} \delta^2[\partial^2\bar{B} - \bar{\partial}^2 B]e^{i\frac{k}{4\pi}\int dt \int_{\Sigma} d^2x(B\partial_t\bar{B} - \bar{B}\partial_t B) - i\frac{k}{4\pi}\int_{\Sigma} d^2x\bar{B}\,b - i\frac{k}{2\pi}\int_{\Sigma} d^2x A^{(0)}\,b - i\frac{k}{4\pi}\int_{\Sigma} d^2x A^{(0)}\bar{A}^{(0)}}. \tag{36}$$

This functional is not physical because it is gauge-variant and so we will project onto the physical wave-functional following [30]. If $U_{\alpha}$ is the unitary operator implementing $U_{\alpha}\hat{b}U_{\alpha}^{-1} = \hat{b}^{(\alpha)} = \hat{b} + \partial^2\alpha$ then[5]

$$(U_{\alpha}\psi)[b] = \langle b|U_{\alpha}|\psi\rangle = \psi[b^{(-\alpha)}] = e^{-i\frac{k}{4\pi}\int_{\Sigma} d^2z\, \partial^2\alpha\bar{\partial}^2\alpha + i\frac{k}{2\pi}\int_{\Sigma} d^2x(A^{(0)}+b)\bar{\partial}^2\alpha + i\mu\sum_a q^a\alpha(\mathbf{z}_a,\bar{\mathbf{z}}_a)}\psi[b]. \tag{37}$$

---

[5]To be explicit, the path-integral over $B_{ij}$ with fixed boundary condition $b^{(-\alpha)}$ that defines $\psi[b^{(-\alpha)}]$ can be converted to a path-integral over $\tilde{B}_{ij} = B_{ij} - (\partial_i\partial_j - \frac{1}{2}\partial^2)\alpha$ (with $\alpha$ time independent) with fixed boundary condition $b$. This is $\psi[b]$ up to the phase we have written.

The physical wave-functional, $\Psi$, is now found by applying the singlet projector $\int \mathcal{D}\alpha\, U_\alpha \circ$ to $\psi[b]$. This is a simple Gaussian path-integral for $\alpha$. Importantly the modes of $\alpha$ annihilated by $\partial_z^2$ and $\partial_{\bar z}^2$

$$\alpha_{zero} = \alpha^{(0)} + \alpha^{(1)} z + \bar\alpha^{(1)} \bar z + \alpha^{(2)} z\bar z\,, \tag{38}$$

do not participate in the Gaussian action and their integrations yield delta functions forcing the total charge, the total dipole moments, and the total trace-quadrupole moment to vanish on the two-surface, $\Sigma$:

$$\int d\alpha^{(0)} d\alpha^{(1)} d\bar\alpha^{(1)} d\alpha^{(2)} \rightarrow \delta\left[\sum_a q^a\right] \delta\left[\sum_a q^a\, \mathbf{z}_a\right] \delta\left[\sum_a q^a\, \bar{\mathbf{z}}_a\right] \delta\left[\sum_a q^a\, \mathbf{z}_a \bar{\mathbf{z}}_a\right]. \tag{39}$$

The Gaussian integration gives us the formal expression:

$$\Psi[b] = \mathcal{N}\left(\delta[\,]'s\right) \times e^{i\frac{k}{4\pi}\int d^2 z \left(\bar\partial^2 (A^{(0)}+b)+\frac{2\pi}{k}\rho\right)(\partial\bar\partial)^{-2}\left(\bar\partial^2(A^{(0)}+b)+\frac{2\pi}{k}\rho\right)} \psi[b]\,, \tag{40}$$

where $\mathcal{N}\left(\delta[\,]'s\right)$ is a shorthand for the delta functions in (39) times possible (field independent) constants from integration.

Let us now use $\Sigma = \mathbb{R}^2$ for which tensor-Gauss law constraint is solved[6] by $B = \partial^2 \phi$ and $\bar B = \bar\partial^2 \phi$ for a single-valued $\phi$. The measure times the delta function is invariant[7]:

$$\left.\frac{\mathcal{D}B_{ij}}{\mathcal{V}_g}\right|_{B[\Sigma]=b} \delta[\partial^2 \bar B - \bar\partial^2 B] = \frac{\mathcal{D}\phi}{\mathcal{V}_g} \delta[\partial^2 \varphi - b]\,, \tag{42}$$

where $\varphi$ is the boundary value of $\phi$. It is easy to check from substituting in (36) that the wave-function is in fact independent of $b$:

$$\Psi[b] = \mathcal{N}\left(\delta[\,]'s\right) \int \frac{\mathcal{D}\phi}{\mathcal{V}_g} \delta\left[\partial^2 \varphi - b\right] e^{i\frac{\pi}{2k}\int d^2 z \rho(\partial\bar\partial)^{-2}\rho}$$

$$= \mathcal{N}\left(\delta[\,]'s\right) e^{i\frac{\mu^2 \pi}{2k}\sum_{ab} q^a (\partial\bar\partial)^{-2}(\mathbf{z}_a, \mathbf{z}_b) q^b}\,. \tag{43}$$

The independence of $\Psi$ on the boundary data, $b$, signals to us that once the zero total charge, zero total dipole moment, and zero total trace-quadrupole moment delta functions are enforced, the wave-functional is a simple phase. Thus the ground state degeneracy on a punctured $\mathbb{R}^2$ is one.

## 2.2 The edge theory

Now let us focus on sourceless path-integral on $\mathbb{R} \times \Sigma$ where $\Sigma \subset \mathbb{R}^2$ possesses a boundary, $Y = \partial\Sigma$ which is diffeomorphic to a circle. As familiar in gauge theories this boundary explicitly breaks invariance under gauge transformations with support on $Y$ and thus we expect to find contributions from "edge-modes" corresponding to these gauge parameters. Much like ordinary Chern-Simons theory, we will find that the theory localizes on $Y$ (this is just a restatement that the theory is gapped in the bulk). The path-integral can then be thought of as

---

[6]Note that Gauss's law implies that $\bar\partial B dz + \partial \bar B d\bar z$ is a closed one-form and so on $\mathbb{R}^2$ (or any contractible subset) can be written as

$$\bar\partial B = \partial \eta_{z\bar z}\,, \qquad \partial \bar B = \bar\partial \eta_{z\bar z}\,, \tag{41}$$

for some $\eta_{z\bar z}$. However these two equations also tell us $\eta_{z\bar z}$ is a total derivative in $z$ as well as in $\bar z$ and so we arrive at $\eta_{z\bar z} = \partial\bar\partial\phi$. For a more general surface, $\Sigma$, there might be more than one solution to the tensor-Gauss law (up to gauge transformation). See for instance, [1] for a discussion on $\Sigma = T^2$.

[7]This follows a similar argument in [30]. Namely, in writing $B = \partial^2 \phi + \beta$ and $\bar B = \bar\partial^2 \phi + \bar\beta$ where $\partial^2 \bar\beta - \bar\partial^2 \beta \neq 0$, the Jacobian encountered from converting $\mathcal{D}B_{ij} = \mathcal{D}\phi\mathcal{D}\delta B_{ij}$ is formally $\det|\partial^2|^2$ which cancels the inverse determinant from writing $\delta[\partial^2 \bar B - \bar\partial^2 B]$ as $\delta[\beta_{ij}]$.

a "transition-amplitude" of the edge-mode theory. We will return to this interpretation when we discuss entanglement in section 5.

To begin we will treat $A_0$ as a Lagrange multiplier, and we will set its boundary condition as

$$A_0|_Y = 0,\tag{44}$$

which is consistent with the variational principle.[8] The integration over $A_0$ enforces the constraint (25) which we again satisfy by writing

$$A_{ij} = \partial_i \partial_j \varphi - \frac{1}{2}\delta_{ij}\partial^2 \varphi.\tag{46}$$

The remaining action is then a total derivative and so we arrive at a boundary action for $\varphi$

$$S_{tCS} \xrightarrow[A_0 \text{ out}]{\text{integrate}} S_\partial := -\frac{k}{4\pi}\int_{\mathbb{R}\times Y} dt\, \delta^{ij}\partial_t \partial_i \varphi\, \mathbf{d}(\partial_j \varphi),\tag{47}$$

where $\mathbf{d} = dx^i \partial_i$ is the spatial exterior derivative (we have left the pullback to $Y$ notationally implicit). Let us measure the geodesic length along $Y$ with a coordinate called $s$; likewise, let us pick a coordinate $n$ such that $\partial_n$ is normal to $Y$. We then can write this as

$$S_\partial = -\frac{k}{4\pi}\int_{\mathbb{R}\times Y} dt\, ds\left(\partial_t(\partial_n \varphi)\partial_s(\partial_n \varphi) + \partial_t \partial_s \varphi\, \partial_s^2 \varphi\right).\tag{48}$$

Once pulled back to $Y$, $\partial_n \varphi|_Y$ is an independent degree of freedom from $\varphi|_Y$ or $\partial_s \varphi|_Y$. The physics interpretation of these independent degrees of freedom are clear: $\partial_n \varphi$ and $\partial_s \varphi$ should be thought of boundary localized *dipoles* oriented normal and parallel (respectively) to $Y$.

Let us define $\xi := \partial_n \varphi|_Y$. Then $S_{tCS}$ becomes a sum of a chiral scalar and a $z = 3$ *chiral Lifshitz scalar*:

$$S_\partial = -\frac{k}{4\pi}\int_{\mathbb{R}\times Y} dt\, ds\left(\partial_t \xi \partial_s \xi - \partial_t \varphi\, \partial_s^3 \varphi\right).\tag{49}$$

More generally, the *q-tensor theory* (19) gives rise to a series of Lifshitz edge theories. Indeed $A_0$ continues to act as a Lagrange multiplier enforcing

$$\varepsilon^{i_1 j_1}\delta^{i_2 j_2}\ldots\delta^{i_q j_q}\partial_{i_1}\partial_{i_2}\ldots\partial_{i_q}A_{j_1 j_2 \ldots j_q} = 0,\tag{50}$$

which we can solve as

$$A_{i_1 i_2 \ldots i_q} = \partial_{i_1}\partial_{i_2}\ldots\partial_{i_q}\varphi - \text{traces}.\tag{51}$$

The action again pulls back to $Y$:

$$S_{q-tCS} = -\frac{k}{4\pi}\int_{\mathbb{R}\times Y} dt\, ds\, \delta^{i_2 j_2}\ldots\delta^{i_q j_q}\partial_{i_2}\ldots\partial_{i_q}\partial_t \varphi\, \partial_s \partial_{j_2}\ldots\partial_{j_q}\varphi$$

$$= -2^{q-2}\frac{k}{4\pi}\int_{\mathbb{R}\times Y} dt\, ds\, \partial_t \partial_s^{q-1}\varphi\, \partial_s^q \varphi - 2^{q-2}\frac{k}{4\pi}\int_{\mathbb{R}\times Y} dt\, ds\, \partial_t \partial_n \partial_s^{q-2}\varphi\, \partial_s^{q-1}\partial_n \varphi,\tag{52}$$

---

[8]A variation of $S_{tCS}$ yields the boundary term

$$\delta S_{tCS}|_{o.s.} = \frac{k}{4\pi}\int_{\mathbb{R}\times Y} dt\, dx^i \left(A_0 \delta^{jk}\partial_j \delta A_{ik} - \mathbf{d}A_0\, \partial_n \delta A_{in}\right).\tag{45}$$

Fixing $A_0|_Y = 0$ is enough to make this boundary variation stationary.

where in the second line we've used the symmetric trace-less condition to exchange $\partial_n^2$ for $-\partial_s^2$ to reduce the theory to two independent terms[9] with a fixed even and odd number of $\partial_n$ derivatives on $Y$. The $2^{q-2}$ comes from the combinatorics of this choice:

$$\sum_{i=0}^{\lfloor \frac{q-1}{2} \rfloor} \binom{q-1}{2i} = \sum_{i=1}^{\lfloor \frac{q}{2} \rfloor} \binom{q-1}{2i-1} = 2^{q-2}. \tag{53}$$

Calling $\xi = \partial_n \varphi|_Y$ we find the boundary action is again that of two chiral Lifshitz scalars:

$$S_{q-tCS} = \frac{k_{z_1}}{4\pi} \int dt \int_Y ds\, \partial_t \varphi \partial_s^{z_1} \varphi + \frac{k_{z_2}}{4\pi} \int dt \int_Y ds\, \partial_t \xi \partial_s^{z_2} \xi, \tag{54}$$

with critical exponents $z_1 = 2q-1$ and $z_2 = 2q-3$ and couplings $k_{z_1} = (-1)^{q-1} 2^{q-2} k$ and $k_{z_2} = (-1)^{q-2} 2^{q-2} k$. We have integrated by parts frivolously, ignoring any possible "windings" of $\varphi$ but in section 5.2 we quantize the chiral Lifshitz theory more carefully and investigate its thermal partition function which provides an alternate characterization of the ground state entanglement entropy in section 5.

## 3 Fractonic physics and extended operators

Let us now return to the theory on $\mathbb{R}_t \times \mathbb{R}^2$. Much like ordinary Chern-Simons theory, in this theory there are no *local* gauge invariant operators. There are *non-local* or *extended* operators that we can construct, however. The most familiar extended object is a line operator[10] that we will call the *charge defect operator*:

$$\mathbf{L}_q(\vec{\mathbf{x}}) = \exp\left( i\mu q \oint_{\mathcal{C}_t} dt\, A_0(t, \vec{\mathbf{x}}) \right). \tag{55}$$

We are using a relaxed notation $\oint_{\mathcal{C}_t} dt$ to indicate that we allow contours, $\mathcal{C}_t$, to either be infinite in extent or compact (if considering the theory at finite temperature, say). It is important however that $\mathcal{C}_t$ is locked at the spatial point, $\vec{\mathbf{x}}$, by gauge invariance. The inclusion of this operator modifies the tensor-Gauss law (25)

$$\varepsilon^{ij} \delta^{kl} \partial_i \partial_k A_{jl} = \mu \frac{2\pi q}{k} \delta^2(\vec{x} - \vec{\mathbf{x}}) \tag{56}$$

to include a topological defect. In addition to the charge defect operator, we have a second line operator that we will call the *monopole string operator*:

$$\mathbf{M}_p[\mathcal{C}_s] = \exp\left( \frac{i}{2} \mu^{-1} p \oint_{\mathcal{C}_s} ds^i\, \delta^{kl} \partial_k A_{il} \right), \tag{57}$$

where $\mathcal{C}_s$ is a closed spatial contour embedded in $\Sigma$ by $s \to \{\bar{x}^i(s)\}$ and $ds^i = ds \frac{\partial \bar{x}^i}{\partial s}$. It is clear that if $\mathcal{C}_s$ is contractible then (57) can be trivially rewritten as $\exp\left( i\mu^{-1} p \int_{\Sigma_{\mathcal{C}_s}} d^2x\, \varepsilon^{ij} \delta^{kl} \partial_i \partial_k A_{jl} \right)$

---

[9]At first glance, $A_{i_1 i_2 \ldots i_q}$ seems to have potentially more degrees of freedom than the two-index theory. However symmetric traceless tensors are extremely constrained in two dimensions. For instance in the $(z, \bar{z})$ coordinates with metric $\delta_{ij} dx^i dx^j = dz d\bar{z}$, the only non-zero components are $A_{zz\ldots z}$ and $A_{\bar{z}\bar{z}\ldots\bar{z}}$.

[10]In truth, since we are always quantizing the theory on constant-time surfaces, $\mathbf{L}_q$ is not an operator in the sense of a map from the Hilbert space to itself. Instead we simply mean an object to be inserted into the path-integral. Similarly all manipulations in this section should be thought of as taking place inside the path-integral.

where $\Sigma_{\mathcal{C}_s}$ is the open two-surface bounded by $\mathcal{C}_s$ and so the expectation value of (57) in any gauge invariant state is 1 by the usual tensor-Gauss law constraint. However, if $\Sigma_{\mathcal{C}_s}$ is pierced by any charge defect lines then the modification of the tensor-Gauss law constraint, (56), implies that $\mathbf{M}_p$ counts the number of defects wrapped by $\mathcal{C}_s$, e.g.

$$\langle \mathbf{M}_p[\mathcal{C}_s] \rangle = \exp\left( i\frac{\pi}{k} \sum_{a \in \Sigma_{\mathcal{C}_s}} q^a \, p \right). \tag{58}$$

The expectation value of $\mathbf{M}$ in any gauge invariant state is invariant under deformations of $\mathcal{C}_s$ that do not cross any charge defects and so from here on we will drop the dependence of $\mathbf{M}$ on the contour unless specifically needed.

A third class of gauge invariant line operators are *dipole string operators* labelled by a spatially constant vector field, $\vec{\mathbf{v}}$, and a spatial closed curve $\mathcal{C}_s$:

$$\mathbf{D}_{\vec{\mathbf{v}}}[\mathcal{C}_s] = \exp\left( i\oint_{\mathcal{C}_s} ds^i \mathbf{v}^j A_{ij}(s) - i\oint_{\mathcal{C}_s} ds\, \mathbf{v} \cdot \bar{x}(s) \varepsilon^{ij} \partial_i A_{jk} \hat{n}_{\mathcal{C}_s}^k \right). \tag{59}$$

Here $\hat{n}_{\mathcal{C}_s}$ is the unit normal vector to $\mathcal{C}_s$ within $\Sigma$. The particular combination in the exponent of (59) can indeed be rewritten as

$$\mathbf{D}_{\vec{\mathbf{v}}}[\mathcal{C}_s] = \exp\left( i\int_{\Sigma_{\mathcal{C}_s}} d^2 x\, \vec{\mathbf{v}} \cdot \vec{x}\, \varepsilon^{jk} \delta^{lm} \partial_j \partial_l A_{km} \right), \tag{60}$$

where again, $\Sigma_{\mathcal{C}_s}$ is the two-surface bounded by $\mathcal{C}_s$ and makes its gauge invariance manifest. This also indicates via (56) that $\mathbf{D}_{\vec{\mathbf{v}}}$ measures the total dipole moment (in the $\mathbf{v}$ direction) of charge defects, $\{q^a\}$, bounded by $\mathcal{C}_s$:

$$\langle \mathbf{D}_{\vec{\mathbf{v}}}[\mathcal{C}_s] \rangle = \exp\left( -i\frac{2\pi}{k}\mu \sum_a q^a \vec{\mathbf{v}} \cdot \vec{\mathbf{x}}_a \right) \tag{61}$$

and that $\langle \mathbf{D}_{\vec{\mathbf{v}}}[\mathcal{C}_s] \rangle$ is invariant under deformations of $\mathcal{C}_s$ that do not cross any defects.

Lastly we have a set of gauge invariant *trace-quadrupole* string operators

$$\mathbf{T}_\nu[\mathcal{C}_s] = \exp\left( i\mu\nu \oint_{\mathcal{C}_s} ds^i\, \bar{x}^j(s) A_{ij}(s) - i\frac{\mu\nu}{2}\oint_{\mathcal{C}_s} ds\, \bar{x}(s)^2 \varepsilon^{ij} \partial_i A_{jk} \hat{n}_{\mathcal{C}_s}^k \right), \tag{62}$$

which by an argument wholly similar to that of the monopole and dipole string operators measures the trace-quadrupole moment of the charge defects bounded by $\mathcal{C}_s$:

$$\langle \mathbf{T}_\nu[\mathcal{C}_s] \rangle = \exp\left( -i\frac{\pi}{k}\mu^2 \sum_a q^a \mathbf{x}_a^2 \right) \tag{63}$$

and is invariant under deformations of $\mathcal{C}_s$. It is important to mention that on $\mathbb{R}^2$ the operators $\{\mathbf{M}_p, \mathbf{D}_{\vec{\mathbf{v}}}, \mathbf{T}_\nu\}$ are not fully independent: indeed, the set of expectation values of $\mathbf{M}_p[\mathcal{C}_s]$ over all possible curves establishes the local positions of all charge defects and fixes the expectation values of $\mathbf{D}_{\vec{\mathbf{v}}}$ and $\mathbf{T}_\nu$.

Now let us comment on the fractonic physics of the charge-defect line operators, $\mathbf{L}_q[\mathcal{C}_t]$. In particular, we noted that gauge invariance requires that $\mathcal{C}_t$ is a constant-space contour. This is essentially a restatement that dipole moment conservation locks the positions of charges. This then suggests that this defect can be partially "freed" by the inclusion of an additional defect.

To see how this works in the present case, suppose we take the product of two "locked-in" charge defects at separated points $\vec{\mathbf{x}}_1$ and $\vec{\mathbf{x}}_2$:

$$\mathbf{L}_q \mathbf{L}_{q'} = \exp\left( i\mu q \oint_{\mathcal{C}_t} dt\, A_0(t, \vec{\mathbf{x}}_1) + i\mu q' \oint_{\mathcal{C}_t} dt\, A_0(t, \vec{\mathbf{x}}_2) \right). \tag{64}$$

To be clear, we are taking this product within the path-integral; that is to say these statements hold as expectation values in gauge-invariant states. We find a interesting effect if we tune $q' = -q$. In this case we can choose a constant-time path $\mathcal{C}_\sigma$ running from $\vec{\mathbf{x}}_2$ to $\vec{\mathbf{x}}_1$ parameterized by $\bar{x}^i(\sigma)$ with $\sigma \in (0, 1)$ and rewrite (64) trivially as

$$\mathbf{L}_q \mathbf{L}_{-q} = \exp\left( i\mu q \oint_{\mathcal{C}_t} dt \int_0^1 d\sigma\, \frac{\partial \bar{x}^i}{\partial \sigma} \partial_i A_0 \right). \tag{65}$$

Now we are allowed to use $A_{ij}$ to deform $\mathcal{C}_t$ to a contour $\mathcal{C}_\eta = \{\bar{t}(\eta), \bar{x}^j(\eta)\}$ in a direction locally orthogonal to $\mathcal{C}_\sigma$ and still maintain gauge invariance:

$$\exp\left( i\mu q \oint_{\mathcal{C}_t} dt \int_0^1 d\sigma\, \frac{\partial \bar{x}^i}{\partial s} \partial_i A_0 \right) \sim \exp\left( i\mu q \oint_{\mathcal{C}_\eta} d\eta \int_0^1 d\sigma\, \left( \frac{\partial \bar{t}}{\partial \eta} \frac{\partial \bar{x}^i}{\partial \sigma} \partial_i A_0 + \frac{\partial \bar{x}^i}{\partial \eta} \frac{\partial \bar{x}^j}{\partial \sigma} A_{ij} \right) \right), \tag{66}$$

(where by "$\sim$" we mean possessing equal expectation value in gauge invariant states)[11] as depicted in figure 1. We will call this new object $\mathbf{S}_q[\bar{t}, \vec{\bar{x}}]$. The orthogonal deformability of $\mathbf{L}_q \mathbf{L}_{-q} \to \mathbf{S}_q$, only allowed after the inclusion of the second charge defect, should be interpreted as the transverse mobility of the dipole formed by $(q, -q)$. A similar phenomenon was noted of the extended operators in [14]: "strips" extended in the $\hat{x}$ direction could be freely deformed in the $\hat{y}$ direction and vice-versa. This theory has *strip-operators* associated to dipoles in any direction (reflecting the continuum rotational symmetry), however their mobility is restricted by the trace-quadrupole moment.

Now let us discuss the strip operators allowed by gauge-invariance a bit more generically. Let $\mathcal{I}$ be a strip parameterized by two "world-sheet" coordinates $(\eta, \sigma)$ with the requirement that $\sigma$ is a compact coordinate that takes values (w.l.o.g.) between 0 and 1 and $\mathcal{I}$ has no boundary in $\eta$. Embedding $\mathcal{I}$ into $\mathbb{R}_t \times \mathbb{R}^2$ with embedding functions $\{\bar{t}(\eta, \sigma), \bar{x}^i(\eta, \sigma)\}$, we define a corresponding strip-operator as

$$\mathbf{S}_q[\mathcal{I}] := \exp\left\{ i\mu q \oint d\eta \int_0^1 d\sigma \left( \left( \frac{\partial \bar{t}}{\partial \eta} \frac{\partial \bar{x}^i}{\partial \sigma} + \frac{\partial \bar{t}}{\partial \sigma} \frac{\partial \bar{x}^i}{\partial \eta} \right) \partial_i A_0 + \frac{\partial \bar{x}^{(i}}{\partial \eta} \frac{\partial \bar{x}^{j)}}{\partial \sigma} A_{ij} \right) \right\}, \tag{69}$$

and demand invariance under (13). After a gauge transformation we have

$$\delta_\alpha \log \mathbf{S}_q = i\mu q \oint d\eta \int_0^1 d\sigma \left( \frac{\partial \bar{t}}{\partial \eta} \frac{\partial \bar{x}^j}{\partial \sigma} \partial_t \partial_j \alpha + \frac{\partial \bar{t}}{\partial \sigma} \frac{\partial \bar{x}^j}{\partial \eta} \partial_t \partial_j \alpha + \frac{\partial \bar{x}^i}{\partial \eta} \frac{\partial \bar{x}^j}{\partial \sigma} \partial_i \partial_j \alpha - \frac{1}{2} \delta_{ij} \frac{\partial \bar{x}^i}{\partial \eta} \frac{\partial \bar{x}^j}{\partial \sigma} \partial^2 \alpha \right). \tag{70}$$

---

[11]The equivalence under orthogonal deformations can be understood from the following: let us imagine deforming $\mathcal{C}_\eta$ to $\mathcal{C}'_\eta$ infinitesimally and let $\tilde{\mathcal{C}}$ be closed curve concatenating $\mathcal{C}_\eta$ with $-\mathcal{C}'_\eta$. Then if the deformation is orthogonal we can write

$$\mathbf{S}[\bar{t}, \bar{x}] \mathbf{S}^{-1}[\bar{t}', \bar{x}'] = \exp\left( i\mu q \int_0^1 d\sigma \oint_{\tilde{\mathcal{C}}} \omega(\sigma) \right) = \exp\left( i\mu q \int_0^1 d\sigma \int_{\Sigma_{\tilde{\mathcal{C}}}} d\omega(\sigma) \right), \tag{67}$$

where $\omega(\sigma) = \frac{\partial \bar{x}^i}{\partial \sigma} \left( \partial_i A_0 dt + A_{ij} dx^j \right)$ and $\Sigma_{\tilde{\mathcal{C}}}$ is the two-surface bounded by $\tilde{\mathcal{C}}$ in a fixed $\sigma$ plane. Pulled back to a fixed $\sigma$ surface, $\Sigma_{\tilde{\mathcal{C}}}$, $d\omega$ vanishes by the constraint generated by $A_{ij}$'s equation of motion:

$$\mathbf{S} \mathbf{S}'^{-1} = \exp\left( i\mu q \int_0^1 d\sigma \int_{\Sigma_{\tilde{\mathcal{C}}}} \frac{\partial \bar{x}}{\partial \sigma} \left( -\partial_i \partial_j A_0 + \partial_t A_{ij} \right) dt \wedge dx^j \right) = 1. \tag{68}$$

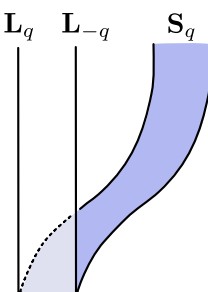

Figure 1: Two charge defects, $\mathbf{L}_q$ and $\mathbf{L}_{-q}$, forming a dipole can be deformed to a strip operator, $\mathbf{S}_q$ as long as that deformation is orthogonal to the dipole moment.

The fourth term gives the necessary condition that if the strip makes an excursion along a spatial direction, then it must do so in a manner locally orthogonal to the dipole directional:

$$\frac{\partial \bar{x}^i}{\partial \eta} \delta_{ij} \frac{\partial \bar{x}^j}{\partial \sigma} = 0. \tag{71}$$

Pulling out a total $\eta$ derivative from the remaining terms we are left with

$$\delta \log \mathbf{S}_q = \exp\left( -i\mu q \oint d\eta \int_0^1 d\sigma \frac{\partial^2 \bar{x}^i}{\partial \eta \partial \sigma} \partial_i \alpha + \frac{\partial^2 \bar{t}}{\partial \eta \partial \sigma} \partial_t \alpha + \frac{\partial \bar{t}}{\partial \sigma} \frac{\partial \bar{t}}{\partial \eta} \partial_t^2 \alpha \right). \tag{72}$$

Gauge invariance requires the vanishing of the exponent of (72) for arbitrary gauge parameters, $\alpha$. We find two large classes[12] of gauge-invariant strip operators roughly distinguished on whether the non-compact parameter $\eta$ is time-like or space-like, which we call **Type T** and **Type S**, respectively[13]

$$
\begin{aligned}
\textbf{Type T}: \quad & \left\{ \bar{t}(\eta), \ \bar{x}^i = X_1^i(\eta) + X_2^i(\sigma) \right\}, \\
\textbf{Type S}: \quad & \left\{ \bar{t}(\sigma), \ \bar{x}^i = F(\sigma) X_1^i(\eta) + X_2^i(\sigma) \right\},
\end{aligned} \tag{73}
$$

supplemented with the orthogonality constraint (71). We note that **Type T** strip operators contain the example constructed at the beginning of this section: a pair of charge defects deformed orthogonally to their dipole moment. **Type S** embeddings include strip operators localized on a time-slice which are a large generalization of the strip operators described in [14]. We illustrate a few examples of **Type T** and **Type S** strip operators in figure 2.

Lastly let us briefly discuss composite strip operators. Unlike line operators, there are multiple ways two strip operators can form a composite strip operator as we depict in figure 3. The first involves overlapping two strips operators (with possibly different charges, $q_1$ and $q_2$) on the same strip embedding, $\mathcal{I}$. The composite is a strip on $\mathcal{I}$ with the sum of the charges, $q_1 + q_2$,

$$\textbf{Fusion:}, \qquad \mathbf{S}_{q_1}[\mathcal{I}] \mathbf{S}_{q_2}[\mathcal{I}] \sim \mathbf{S}_{q_1+q_2}[\mathcal{I}], \tag{74}$$

as depicted in figure 3a. We will call this *fusion* because of the obvious analog to the Abelian fusion of Wilson lines. The second case involves the product of strip operators of identical charge, $q$, and with strips, $\mathcal{I}_1$ and $\mathcal{I}_2$ touching adjacent to their dipole moment. As depicted in figure 3b, the composite is a strip operator of the same charge but whose embedding is the concatenation, $\mathcal{I}_1 \cup \mathcal{I}_2$, of the strips, an equivalence we will call *zippering*:

$$\textbf{Zippering:}, \qquad \mathbf{S}_q[\mathcal{I}_1] \mathbf{S}_q[\mathcal{I}_2] \sim \mathbf{S}_q[\mathcal{I}_1 \cup \mathcal{I}_2]. \tag{75}$$

Again these statements are taken to hold in expectation values of gauge invariant states.

---

[12]These two classes are likely not exhaustive.

[13]By convention we will take $\bar{t} =$constant to be **Type S**.

# 4 Large gauge transformations, dipole quantization, and dipole condensation.

Up to this point we have only required invariance under (13), essentially regarding the gauge group as $\mathbb{R}$. For this non-compact symmetry are no "large gauge" transformations (we clarify this terminology shortly) and there is no quantization of the level, $k$, or the charges associated to any line, string, or strip operator. In this section we ask what circumstances allow us to impose invariance under a compact group, $U(1)$, and what consequences does this invariance

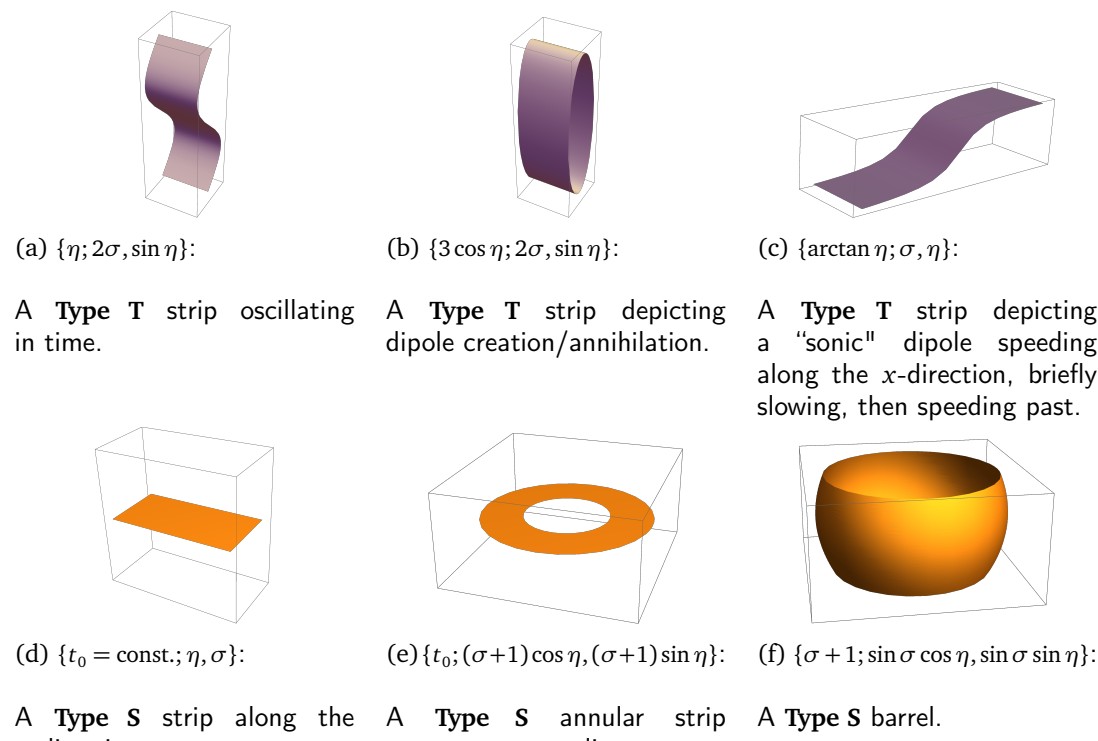

(a) $\{\eta; 2\sigma, \sin\eta\}$:

A **Type T** strip oscillating in time.

(b) $\{3\cos\eta; 2\sigma, \sin\eta\}$:

A **Type T** strip depicting dipole creation/annihilation.

(c) $\{\arctan\eta; \sigma, \eta\}$:

A **Type T** strip depicting a "sonic" dipole speeding along the $x$-direction, briefly slowing, then speeding past.

(d) $\{t_0 = \text{const.}; \eta, \sigma\}$:

A **Type S** strip along the $x$-direction on a constant $t$ slice.

(e) $\{t_0; (\sigma+1)\cos\eta, (\sigma+1)\sin\eta\}$:

A **Type S** annular strip on a constant $t$ slice.

(f) $\{\sigma+1; \sin\sigma\cos\eta, \sin\sigma\sin\eta\}$:

A **Type S** barrel.

Figure 2: Three different gauge-invariant **Type T** strip operators (in purple) and three different gauge-invariant **Type S** strip operators (in orange). Time runs upward in these figures. The parameterizations generating the figures are provided in the form $\{\bar{t}; \bar{x}^i\}$. It is easy to verify each parameterization matches either **Type T** or **Type S** in (73) and satisfies the orthogonality constraint, (71).

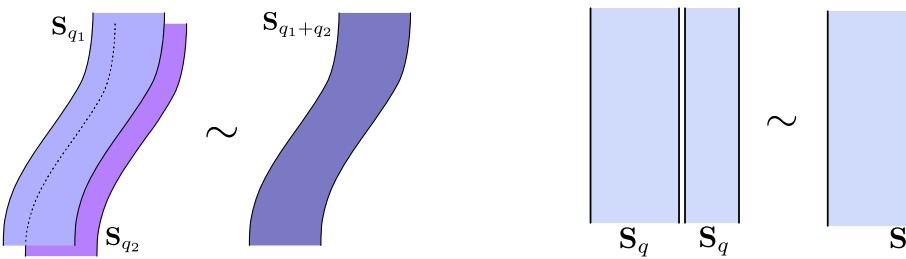

(a) The fusion of two strips to form a composite strip with charge $q_1 + q_2$.

(b) Two strips (of the same charge) adjacent along their "dipole moment" can "zipper" into wider strip operator (of the same charge).

Figure 3: Two possible ways strip operators can form a composite strip operator.

have for the charges and the vacuum of the theory.

To begin we first must state clearly what is meant by associating the group $U(1)$ to (13): there exists a unique association of gauge parameters to group elements compatible with the group structure:

$$\alpha(t,\vec{x}) \rightarrow g_\alpha(t,\vec{x}) \in U(1), \qquad g_\alpha g_\beta = g_{\alpha+\beta}, \qquad g_\alpha^* = g_{-\alpha}. \qquad (76)$$

Although this statement is basic, we already see the first subtlety in defining a compact group structure for tensor gauge theories, namely because $\alpha$ has engineering dimensions of length, *this map must involve a fundamental (inverse) length scale*. This is the origin of $\mu$ appearing in the previous sections:

$$g_\alpha(t,\vec{x}) = \exp\left(i\mu\,\alpha(t,\vec{x})\right). \qquad (77)$$

So far there has been no conspiracy: we have simply been choosing to measure charges of line and string operators in units of $\mu$. However we shall soon see a physical motivation for this choice.

We can now consider *large gauge transformations*: single valued group elements, $g_\alpha(t,\vec{x})$, with single-valued field variations, $\{\delta_\alpha A_0, \delta_\alpha A_{ij}\}$, but with parameters, $\alpha(t,\vec{x})$, that are not single-valued. One such gauge transformation arises when considering the theory on a Euclidean thermal circle of radius $\beta$:

$$\alpha(\tau,\vec{x}) = \mu^{-1}\frac{2\pi m\,\tau}{\beta}, \qquad g_\alpha = e^{i\frac{2\pi m\tau}{\beta}}, \qquad m \in \mathbb{Z}. \qquad (78)$$

Invariance under these gauge transformations then imply that charge defect operators have *integer charges* (measured in units of $\mu$)

$$\mathbf{L}_q[A_0 + \delta_\alpha A_0] = \mathbf{L}_q[A_0]\exp\left(i2\pi mq\right) \qquad \Rightarrow \qquad q \in \mathbb{Z}. \qquad (79)$$

There is no other quantization stemming from this large gauge transformation (in particular $\delta_\alpha A_{ij} = 0$ under (78)). Allowing ourselves to "puncture" $\mathbb{R}^2$, we can also consider spatial large gauge transformations of the form

$$\alpha(t,\vec{x}) = \mu^{-1}m\,\theta, \qquad g_\alpha = e^{im\theta},, \qquad m \in \mathbb{Z}, \qquad (80)$$

where $\theta$ is a polar angle around the puncture point, $\vec{x}_o$. In appendix B we show that under (80), the "flux" picks up a contact contribution at $\vec{x}_o$:

$$\varepsilon^{ij}\delta^{kl}\partial_i\partial_k\left(\delta_\alpha A_{jl}\right) = \mu^{-1}\pi m\,\partial^2\delta^2(\vec{x}-\vec{x}_o). \qquad (81)$$

While $\mathbf{M}_p$ and $\mathbf{D}_{\vec{v}}$ are invariant under (80), $\mathbf{T}_\nu$ transforms and so its charge must be quantized:

$$\mathbf{T}_\nu[A_{ij} + \delta_\alpha A_{ij}] = \mathbf{T}_\nu[A_{ij}]\exp\left(-i2\pi\nu\right) \qquad \Rightarrow \qquad \nu \in \mathbb{Z}. \qquad (82)$$

This begs the question: are there large gauge transformations that quantize $\mathbf{M}_p$ and $\mathbf{D}_{\vec{v}}$? Because $\mathbf{D}_{\vec{v}}$ has a unitless "charge," $\vec{v}$, and $\mathbf{M}_p$ has a charge measured in units of $\mu^{-1}$, it clear that extra dimensionful factors must come into play in $\alpha$ in order for this to work. We have natural candidates suggested by the *global charges* (for which $\delta_\alpha A_{ij} = \delta_\alpha A_0 = 0$ exactly):

$$\alpha = \mu^{-1}\Lambda^{(0)} + \Lambda_i^{(1)} x^i + \mu\Lambda^{(2)}x^2, \qquad (83)$$

where $\Lambda^{(0,1,2)}$ are dimensionless constants. We can think of (80) as a gauging of $\Lambda^{(0)}$ to $\lambda^{(0)}(t,\vec{x}) = m\theta$. Is there a sense in which we can gauge, say, $\Lambda_i^{(1)} \rightarrow \lambda_i^{(1)}(t,\vec{x}) = m\theta$ such that

it winds around a spatial cycle? The obvious hangup is that the associated group element is *prima facie* not single-valued under such a gauge parameter:

$$g_{\lambda_i^{(1)} x^i} = e^{im\theta \mu x^i}. \tag{84}$$

Let us amuse ourselves with forcing the above to be single-valued by requiring $\mu x^i \in \mathbb{Z}$. That is we imagine that positions secretly lie in a lattice[14] with spacing $\mu^{-1}$. This may seem like a perverse course of action: we began with a theory in the continuum and it is legitimate to concern ourselves with a gauge symmetry that is defined strictly in the continuum. We discuss this avenue briefly in the "$U(1)$ **vs.** $\mathbb{R}$" portion of the discussion (6) where we recast our interpretation of the theory and its entanglement entropy for the non-compact symmetry. However it is also not uncommon for a "fractonic field theory" to retain some memory of an underlying lattice e.g. in order to assign ground state degeneracies [14–16], to regulate the energies of states charged under subsystem symmetries [14–16], and to fix hydrodynamic Ward identities [31]. The common thread in these examples is the key role the lattice plays in organizing symmetry and its subsequent influence on universal aspects of the continuum theory. Below we outline some motivations for regarding the gauge symmetry of this model, emergent at low energies, as organized by an underlying lattice.

- Firstly, we recall from section 1 that (9) can be viewed as a relevant deformation of the low-energy tensor gauge theory describing the rotationally symmetric point of the XY-plaquette model on a square lattice, $\mathfrak{L}$:

$$H = \sum_{\hat{r} \in \mathfrak{L}} \left( \pi_{\hat{r}} \pi_{\hat{r}} - K \cos\left(\hat{\Delta}_{xy} \phi_{\hat{r}}\right) - \frac{K}{2} \cos\left(\hat{\Delta}_{xx} \phi_{\hat{r}}\right) - \frac{K}{2} \cos\left(\hat{\Delta}_{yy} \phi_{\hat{r}}\right) \right), \tag{85}$$

  where the gauge theory arises from the gauging of the shift symmetry $\phi \to \phi + \alpha$. Because $\phi$ appears as a phase in the Hamiltonian, $\phi$ is only defined up to local $2\pi$ shifts and so this symmetry is *inherently compact*. The global dipolar shift symmetry

$$\alpha_{\hat{r}} = \Lambda_x^{(1)} \hat{r}_x + \Lambda_y^{(1)} \hat{r}_y, \qquad \Lambda^{(1)}{}_{,i} \sim \Lambda_i^{(1)} + 2\pi \tag{86}$$

  is also compact and so states charged under it have integer charges. The continuum limit of (85) is

$$H = \int d^2x \left( \tilde{\pi}(x)\tilde{\pi}(x) + \frac{K}{2} \left( \partial_i \partial_j \tilde{\phi} \right)^2 \right), \tag{87}$$

  where $\tilde{\pi} \sim a\pi$ and $\tilde{\phi} \sim a\phi$ where $a$ is the lattice spacing. We have maintained the units of $[K] \sim \ell^{-1}$. The continuum variable, $\tilde{\phi}$ still possesses a dipolar shift symmetry, $\tilde{\phi} \to \tilde{\phi} + \tilde{\alpha}$ with

$$\tilde{\alpha} = \Lambda_i^{(1)}(a\hat{r}_i) \sim \Lambda_i^{(1)} x^i. \tag{88}$$

  In the strict continuum limit this symmetry is no longer compact (i.e. $g = \exp(ia^{-1}\tilde{\alpha})$ is not single valued under $\Lambda_i^{(1)} \sim \Lambda_i^{(1)} + 2\pi$), but we still might demand organizing the states of this continuum theory by integer charges and effectively regarding it as compact.

- Secondly, while dipoles in the continuum can involve any possible length scale, the interpretation of linearized 2D elasticity [25] as mentioned in section 1 suggests fixing this in units of the lattice spacing. Indeed, an important piece of this dictionary is identification of a dipole excitation with a dislocation and the dipole moment with its Burgers

---

[14]In this case it is a square lattice.

vector. Since the Burgers vector counts the failed closure of a lattice path, it is always quantized in units of the lattice spacing. In allowing lattice quantized dipole moments, it is natural to also consider lattice quantized gauge transformations such as (84). In fact, we shall soon see the effect of such a gauge transformation is to shift dipole moments by an integer in units of the lattice spacing.

Charging unabashedly forward, we will not worry ourselves upon seeing the combination $\mu\vec{x}$ in an exponential, regarding it formally as an integer. Requiring $\delta_\alpha A_{ij}$ to be single-valued, we then have two additional large gauge transformations[15]

$$\alpha_1^i = m\theta\, x^i, \qquad g_{\alpha_1^i} = e^{im\theta\mu x^i} \qquad \text{and} \qquad \alpha_2 = m\theta\,\mu\, x^2, \qquad g_{\alpha_2} = e^{im\theta\mu^2 x^2}, \tag{89}$$

which quantize the charges of $\mathbf{D}_{\vec{\mathbf{v}}}$ and $\mathbf{M}_p$, respectively[16]:

$$\mathbf{v}^i \in \mathbb{Z}, \qquad\qquad p \in \mathbb{Z}. \tag{90}$$

Quantization of dipole moments, which from here on we will take in units of $\mu^{-1}$, implies that the level must be quantized $k \in \mathbb{Z}$ as well, via an argument found in [1]. Re-examining the expectation values of the string operators in light of this quantization

$$\langle\mathbf{M}_p\rangle = e^{i\frac{\pi}{k}p\sum_a q^a}, \qquad \langle\mathbf{D}_{\vec{\mathbf{v}}}\rangle = e^{i\frac{2\pi}{k}\mathbf{v}_j\sum_a q^a\left(\mu\mathbf{x}_a^j\right)}, \qquad \langle\mathbf{T}_\nu\rangle = e^{i\frac{\pi}{k}\nu\sum_a q^a\left(\mu^2\mathbf{x}_a^2\right)}, \tag{91}$$

we find that gauge invariant operators can only distinguish charge defects mod $2k$:

$$\sum_a q^a \sim \sum_a q^a + 2k\mathbb{Z}, \tag{92}$$

and so only $q \in \mathbb{Z}_{2k}$ are physically distinguishable. Another way of stating this is that the action of a large gauge transformation on $\mathbf{M}_{p=1}$ is to shift the total charge by $2k$. Similarly, in addition to quantization of dipole moments, we find that they also form equivalence classes

$$d^i := \sum_a q^a\left(\mu\mathbf{x}_a^j\right) \sim d^i + k\mathbb{Z}, \tag{93}$$

under the action of large gauge transformations. Lastly the trace of the quadrupole moment falls into equivalence classes of $\mathbb{Z}_{2k}$

$$t := \sum_a q^a\left(\mu^2\mathbf{x}_a^2\right) \sim t + 2k\mathbb{Z}. \tag{94}$$

Treating dipoles as the important degrees of freedom, we see that the vacuum forms a *dipole condensate*, allowing dipoles of moment $d^i = k$ to become transparent. We elaborate on this in the next section.

## 4.1 Restored mobility and quantum hall physics

We now briefly explain how condensation of $d^i = k\mathbb{Z}$ dipole moments restores full mobility to charge and dipolar excitations, at least in a microscopic sense. Given a dipole excitation (or generally a Type T strip operator) of moment $q(\mu\ell)\hat{e}^i$, we can imagine introducing a transparent dipole of moment $q\frac{nk}{\gcd(q,k)}\hat{e}^i$ (with $n \in \mathbb{Z}$) adjacent to its dipole moment. By the zippering

---

[15]We also note that there are also transformations of the form $\alpha = m\frac{2\pi\tau}{\beta}x^i$ and $\alpha = m\frac{2\pi\tau}{\beta}\mu x^2$, however these do not provide any additional quantization.

[16]Details can be found in appendix B.

property of strip operators this is equivalent to a dipole of moment $q\left(\mu\ell + n\frac{k}{\gcd(q,k)}\right)\hat{e}^i$. However we can also "zipper off" the same transparent dipole on the opposite side of this composite operator and let it fall into the condensate. The result is that our original dipole has "hopped" $\frac{nk}{\gcd(q,k)}$ lattice units in the direction along its dipole moment. From the macroscopic perspective the condensate has restored full mobility to dipolar excitations. By a similar argument, a charge defect $q$ can "hop" $\frac{nk}{\gcd(k,q)}$ lattice units in any direction by pulling an appropriate dipole out of the condensate. This is depicted in figure 4 below. We mention that in the context of elasticity, the ability to condense dipoles (i.e. dislocations) has been noted by previously, e.g. in [32–34], where it signals a "quantum melting" transition.

We emphasize even though transparent dipoles mediate the hopping of charges, dipoles of moments smaller than $k$ cannot "collapse" and remain robust and distinct excitations of in this model. In fact there is a sense in which we can regard dipolar excitations as fundamental excitations and charges as "composites." Indeed, the net charge on $\mathbb{R}^2$ must vanish and so a single charge must have a partner somewhere: the simplest charge configuration is a (albeit possibly very long) dipole. Due to the dipole condensate we can then always exchange this long dipole for a small collection of separated "fundamental" dipoles (i.e. moments less than $k$) as illustrated in figure 5. As a result of this the charge within a region is not a fixed quantity: charges can be exchanged for dipole moments.

Thus the "long-distance physics" of this model, as summarized in figure 6, is composed of two types of mobile quasi-particles: $(a^x)_{d^x}$ and $(a^y)_{d^y}$ ($d^i \in \mathbb{Z}_k$). We can assign statistics to these quasi-particles by wrapping dipole string operator with charge $\mathbf{v}^i = d_1^i$ around a $d_2^j$ dipolar quasi-particle. It is easy to see from the expectation value (61) that the statistics, $\mathcal{S}$, are anyonic and mutually trivial:

$$\mathcal{S}_{d_1 d_2} = e^{i\frac{2\pi}{k}d_1^i \delta_{ij} d_2^j}. \tag{95}$$

Thus as emphasized in section 1 this tensor gauge theory, with compact gauge group and dipole quantization, can aptly be called a *quantum Hall fluid of dipoles*.

# 5 Entanglement

Now let us bolster the above description by examining the sub-region entanglement of the ground-state. We will consider a contractible subregion, $\mathcal{A} \subset \mathbb{R}^2$, with the topology of a disc

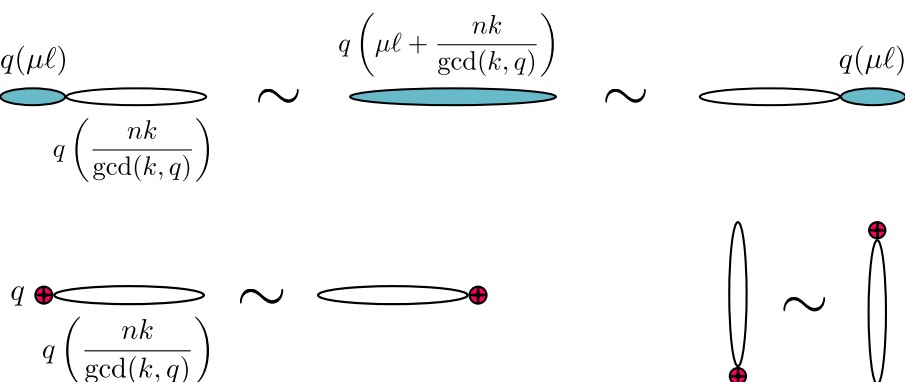

Figure 4: In the top, a dipole (the teal oval) can effectively "hop" in the direction along its dipole moment by merging it with a transparent dipole (i.e. one with dipole moment in $k\mathbb{Z}$) and condensing one from the other end. Below the same mechanism allows charge defects to hop in either direction.

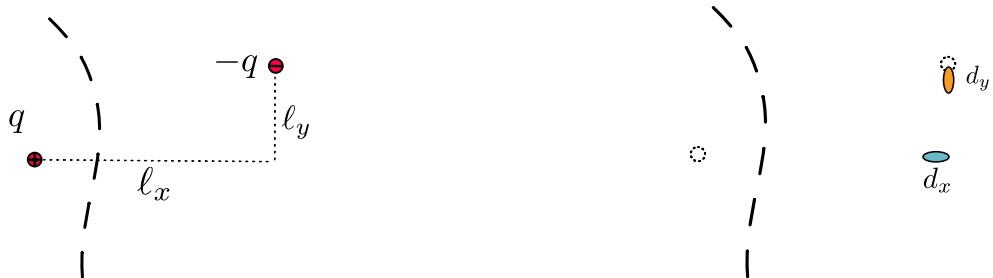

Figure 5: (Left) A net charge to the left of the dashed line has a partner in the complement region. By condensing long transparent dipoles, we can decompose this configuration into two fundamental dipoles with moments $d_{x,y} = q\left(\mu\ell_{x,y} - \lfloor\frac{\mu\ell_{x,y}\gcd(k,q)}{k}\rfloor\frac{k}{\gcd(k,q)}\right)$.

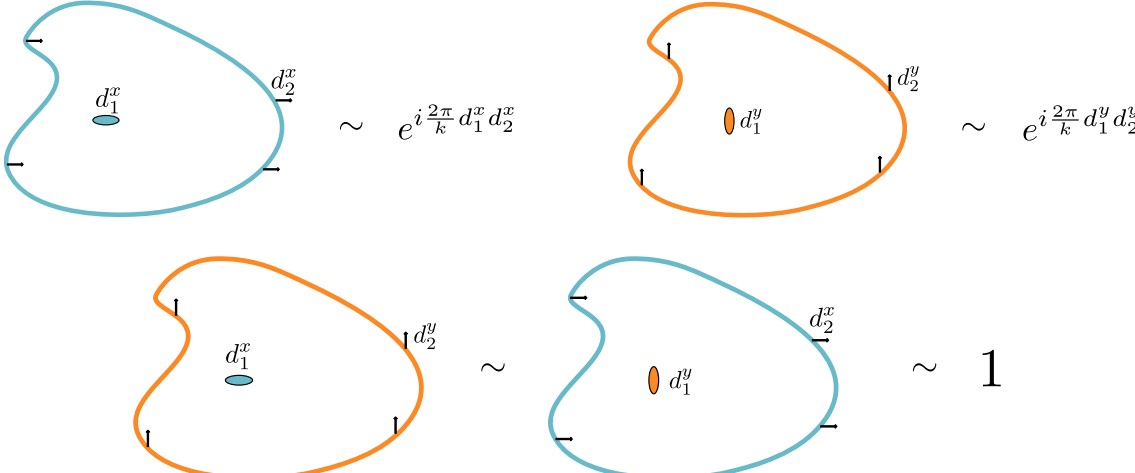

Figure 6: A dipole string operator wrapping around a dipole of the same orientation results in a phase reminiscent of anyon statistics. Dipoles of orthogonal orientations are mutually transparent.

but otherwise geometrically generic. As is well known, the entanglement entropy in any field theory is expected to display divergences scaling with the subsystem size. We are interested in any possible subleading corrections that provide a signature of topological order [6,7]. We will show below that the entanglement entropy in this model, in addition to an ordinary area law, displays a "topological entanglement entropy" consistent with two sets of Abelian topological order comprised of $k$ anyons each. Before diving in completely, let us briefly overview our method here.

## 5.1 The extended Hilbert space

Regarding entanglement, a familiar complication in gauge theories [35–40] is the obstruction of the Hilbert space admitting a tensor-product decomposition along a subregion, $\mathcal{A}$, and its complement, $\mathcal{A}^c$

$$\mathcal{H}_\Sigma \neq \mathcal{H}_\mathcal{A} \otimes \mathcal{H}_{\mathcal{A}^c}. \tag{96}$$

This theory possesses the same complication due to the symmetric traceless gauge symmetry, (13). A simple diagnosis of this problem is that states on the left-hand side of (96) are invariant under *all* gauge transformations, while generic states on the right-hand side are variant under gauge transformations having support on $\partial\mathcal{A}$. Consider the generator of gauge transforma-

tions acting on $\mathcal{H}_\mathcal{A}$ defined by the infinitesimal variation of the action under (13) by a time independent and single-valued $\alpha$ (we will consider the effect of "large gauge transformations shortly)

$$
\begin{aligned}
\hat{\mathcal{Q}}[\alpha] &= -\int d^2x\, \varepsilon^{\langle ij}\delta^{kl\rangle}\partial_i\partial_k\alpha A_{jl} \\
&= -\oint_{\partial\mathcal{A}} ds\, \hat{n}_i \varepsilon^{\langle ij}\delta^{kl\rangle}\partial_k\alpha A_{jl} + \oint_{\partial\mathcal{A}} ds\, \hat{n}_k \varepsilon^{\langle ij}\delta^{kl\rangle}\alpha\,\partial_i A_{jl}\,,
\end{aligned}
\tag{97}
$$

where $s$ coordinatizes $\partial\mathcal{A}$ and $\hat{n}^i$ is the unit outward normal vector to $\partial\mathcal{A}$.[17] $\hat{\mathcal{Q}}$ obeys an Abelian algebra with a central extension:

$$
\left[\hat{\mathcal{Q}}[\alpha], \hat{\mathcal{Q}}[\beta]\right] = i\frac{4\pi}{k}\oint_{\partial\mathcal{A}}\delta^{kl}\partial_k\alpha\,\mathbf{d}(\partial_l\beta)\,.
\tag{99}
$$

This central extension implies that it is not consistent for $\mathcal{H}_A$ to be trivial under gauge transformations. Instead it must be furnished by the representations of this algebra. To understand these representations, let us break down these charges a little bit.

Inside of $\hat{\mathcal{Q}}[\alpha]$ are two independent current algebras associated to the two independent dipole moments and generated by the two independent components of $\partial_i\alpha$ pulled back to $\partial\mathcal{A}$. For instance solving the Gauss law constraint by $A_{ij} = \left(\partial_i\partial_j - \frac{1}{2}\delta_{ij}\partial^2\right)\phi$ we see $Q[\alpha] = \oint \partial_i\alpha\,\mathbf{d}(\partial^i\phi)$. Pulled back to $\partial A$ the gradient of $\alpha$ has the following expansion:

$$
\partial^i\alpha = \sum_{m\in\mathbb{Z}}\lambda_m^i f_m(s)\,,
\tag{100}
$$

where $\{f_m(s)\}_{m\in\mathbb{Z}}$ is a complete orthogonal set of dimensionless functions on $\partial A$. Without loss of generality we will take $s$ to range from $0$ to $\ell$ so that $f_m(s) = e^{i\frac{2\pi m}{\ell}s}$. Calling

$$
\hat{J}_m^i := \mathcal{Q}[\lambda_m^i = 1]\,,
\tag{101}
$$

then (99) separates into two centrally extended chiral $\mathfrak{u}(1)$ algebras:

$$
\left[\hat{J}_m^i, \hat{J}_m^j\right] = \frac{2\pi}{k}(2\pi m)\delta^{ij}\delta_{m+n}\,.
\tag{102}
$$

The $m = 0$ modes commute with all other current modes and will label the representations of these two algebras. These are associated with $\partial^i\alpha = 1$ or $\alpha = (x - x_0)^i$ (at least in a collar neighborhood of $\partial A$). It is easy to see that $\hat{J}_0^i$ is precisely the dipole string operator (59) wrapping $\partial\mathcal{A}$ and measuring the dipole moment in $\mathcal{A}$:

$$
\hat{J}_0^i = -\oint_{\partial\mathcal{A}} ds^j A^i{}_j + \oint_{\partial\mathcal{A}} ds\,\hat{n}^l\,(\bar{x}-x_0)^i \varepsilon^{jk}\partial_j A_{kl} = \frac{2\pi}{k}d_\mathcal{A}^i\,.
\tag{103}
$$

While these eigenvalues are invariant under small gauge transformations (evidenced by $\hat{J}_0^i$ commuting with all other $\hat{J}_{m\neq 0}^i$) the "large gauge transformations" from section 4 shift these eigenvalues by $2\pi\mathbb{Z}$ and so truly gauge invariant states will require a sum over these shifts.

---

[17]We've assumed that the right-hand side of the (25) constraint is zero, but if there are defects, $\{q^a\}$, puncturing $\mathcal{A}$, we can always write $A_{ij} = A_{ij}^{(0)} + B_{ij}$ with $A_{ij}^{(0)}$ a fixed time-independent background configuration satisfying

$$
\varepsilon^{ij}\delta^{kl}\partial_i\partial_k A_{jl}^{(0)} = \mu\frac{2\pi}{k}\sum_a q^a\delta^2(x - \mathbf{x}_a)\,.
\tag{98}
$$

The variation of the action only couples to $B$ and $\hat{\mathcal{Q}}[\alpha]$ has the exact same expression with $A$ replaced with $B$.

To recap, $\mathcal{H}_{\mathcal{A}}$ is spanned by representations of two $\mathfrak{u}(1)$ Kǎc-Moody algebras labelled by the independent components of the dipole moments, and thus $\mathcal{H}_{\mathcal{A}}$ is infinite dimensional. An identical discussion applies for $\mathcal{H}_{\mathcal{A}^c}$. Thus returning to the topic of entanglement, it is clear that $\mathcal{H}_{\Sigma}$, which for $\Sigma = \mathbb{R}^2$ is one-dimensional, cannot be the tensor product of its infinite-dimensional factors. To address this mismatch we will employ a method known as the *extended Hilbert space* [35–37, 40, 41] which is as follows. We map $\mathcal{H}_{\Sigma}$ to a gauge-invariant subspace, $\tilde{\mathcal{H}}_{\Sigma}$, of the gauge-variant $\mathcal{H}_{\mathcal{A}} \otimes \mathcal{H}_{\mathcal{A}^c}$. The prescription for this embedding is obvious: we simply impose gauge invariance by hand. For a physical state $|\psi\rangle$, its image under this embedding $|\tilde{\psi}\rangle \in \tilde{\mathcal{H}} \subset \mathcal{H}_{\mathcal{A}} \otimes \mathcal{H}_{\mathcal{A}^c}$ can then be reduced upon $\mathcal{H}_{\mathcal{A}^c}$ and its entanglement entropy computed. Imposing gauge invariance by hand on $|\tilde{\psi}\rangle \in \tilde{\mathcal{H}}_{\Sigma}$ requires

$$\left( \hat{J}^i_{\mathcal{A},m} \otimes \hat{1}_{\mathcal{A}^c} + \hat{1}_{\mathcal{A}} \otimes \hat{J}^i_{\mathcal{A}^c,-m} \right) |\tilde{\psi}\rangle = 0. \tag{104}$$

The $m = 0$ equations of (104) simply state that the dipole charges in $\mathcal{A}$ and $\mathcal{A}^c$ must be equal and opposite (such that the global state is "dipole neutral"). The $m \neq 0$ equations are more constraining: they enforce the invariance under "small" gauge transformations on $\partial\mathcal{A}$ and are strong enough to project $\mathcal{H}_{\mathcal{A}} \otimes \mathcal{H}_{\mathcal{A}^c}$ to a finite dimensional $\tilde{\mathcal{H}}_{\Sigma}$. These two equations are exactly those defining *Ishibashi states* [42]. We will show that once the dipole charges have been specified, $|\tilde{\psi}\rangle$ has a unique solution that is a tensor product of the two such Ishibashi states[18].

Let us write this state explicitly. Let us suppose that the subsystems, $\mathcal{A}$ and $\mathcal{A}^c$, are pierced with defects totalling to net dipole moments $d^i$ and $-d^i$, respectively. Denoting $|d^i; 0, 0\rangle_{\mathcal{A}}$ as the state in the $d^i$ sector annihilated by positive current modes

$$\hat{J}^i_{\mathcal{A},n} |d^i; 0, 0\rangle_{\mathcal{A}} = 0, \qquad n > 0, \tag{105}$$

then a convenient basis for the $\mathcal{H}_{\mathcal{A}}$ is labelled by occupation numbers of $\hat{J}^i_{\mathcal{A},n<0}$ oscillators:

$$|d^i, \{M^i_m\}\rangle_{\mathcal{A}} := \prod_{i=x,y} \prod_{m=1} \frac{1}{\sqrt{M^i_m!}} \left( \frac{k}{2\pi} \frac{1}{2\pi m} \right)^{M^i_m/2} \left( J^i_{\mathcal{A},-m} \right)^{M^i_m} |d^i; 0, 0\rangle_{\mathcal{A}}. \tag{106}$$

A similar basis exists for $\mathcal{H}_{\mathcal{A}^c}$. The solution of (104) then can be written explicitly as

$$|d^i\rangle\!\rangle = \sum_{z^x,z^y \in \mathbb{Z}} \sum_{\{M^i_m\}=0}^{\infty} \left| d^i + k z^i; \{M^i_m\} \right\rangle_{\mathcal{A}} \otimes \left| -d^i - k z^i; \{M^i_m\} \right\rangle_{\mathcal{A}^c}, \tag{107}$$

where the sums over $z^x$ and $z^y$ arises as the sum over all physically equivalent dipole configurations identified through large gauge transformations. As should be expected from the well-known divergences appearing with Ishibashi states, $|d^i\rangle\!\rangle$, as written above, has infinite norm. To make this a legitimate vector in the Hilbert space it is necessary to smear it in Euclidean time. Recall, however, that all equations of motion in this theory are constraints and so in order to do so, we need to supplement this theory with an auxiliary Hamiltonian. There is no lady of the lake to hand us this Hamiltonian and so we will have to make a choice. We expect this choice to affect the area law but not the universal constant, a point that we elaborate on in appendix A. For the present calculation we will work with the simple sum of current bilinear operators:

$$H_{\mathcal{A}} = \frac{2\pi}{\ell} \sum_{i=x,y} \left( \frac{1}{2} J^i_{\mathcal{A},0} J^i_{\mathcal{A},0} + \sum_{m=1}^{\infty} J^i_{\mathcal{A},-m} J^i_{\mathcal{A},m} \right) \tag{108}$$

---

[18]These states appear generically in extended Hilbert space calculations of entanglement for ordinary Chern-Simons theories [43].

and similarly for $H_{\mathcal{A}^c}$. The total Hamiltonian is then the sum: $\mathbf{H} = \frac{1}{2}\left(H_{\mathcal{A}} \otimes 1_{\mathcal{A}^c} + 1_{\mathcal{A}} \otimes H_{\mathcal{A}^c}\right)$. Then our smeared state

$$|d^i(\varepsilon)\rangle\!\rangle := e^{-\frac{\varepsilon}{2}\mathbf{H}}|d^i\rangle\!\rangle \tag{109}$$

is a normalizable vector.

Let us now trace out the $\mathcal{A}^c$ system from the density matrix $\rho(\varepsilon) = |d^i(\varepsilon)\rangle\!\rangle\langle\!\langle d^i(\varepsilon)|$. The basis states (106) form an $H$ eigenbasis of the either factor of the extended Hilbert space:

$$e^{-\varepsilon H_{\mathcal{A}}}|d^i; \{M_m^i\}\rangle_{\mathcal{A}} = e^{-\frac{2\pi\varepsilon}{\ell}\frac{2\pi^2}{k^2}(\delta_{ij}d^id^j)}\prod_{i=x,y}\left(\prod_{m=1}e^{-\frac{2\pi\varepsilon}{\ell}\frac{2\pi}{k}(2\pi m)M_m^i}\right)|d^i; \{M_n^i\}\rangle_{\mathcal{A}}$$

$$\equiv e^{-\frac{2\pi\varepsilon}{\ell}\frac{2\pi^2}{k^2}(\delta_{ij}d^id^j)}e^{-\varepsilon E_{\{M_n^i\}}}|d^i; \{M_m^i\}\rangle_{\mathcal{A}}. \tag{110}$$

As $|d^i\rangle\!\rangle$ is the maximally mixed state, reducing the density matrix of the smeared $|d^i\rangle\!\rangle(\varepsilon)$ upon $\mathcal{H}_{\mathcal{A}^c}$ results in the mixed density matrix

$$\rho_{\mathcal{A}}(\varepsilon) = \sum_{z^i}\sum_{\{M_m\}}\sum_{\{N_n\}} e^{-\frac{4\pi^3}{k^2}\frac{\varepsilon}{\ell}\delta_{ij}(d^i+kz^i)(d^j+kz^j)}e^{-\varepsilon E_{\{M_m^i\}}}|d^i+kz^i, \{M_m^i\}\rangle_{\mathcal{A}}\langle d^i+kz^i, \{M_m^i\}|_{\mathcal{A}}. \tag{111}$$

A useful physical picture is to view (111) as the thermal density matrix of the edge modes on a regulated entangling surface with inverse temperature proportional to $\varepsilon$, which we explain in more detail in section 5.2. We are interested in the "high-temperature," $\varepsilon \to 0$, limit of its entropy:

$$S_{\mathcal{A}} = \lim_{\varepsilon\to 0}\left(1 - \varepsilon\frac{\partial}{\partial\varepsilon}\right)\log\mathrm{tr}\rho_{mixed}(\varepsilon). \tag{112}$$

Taking the trace we have

$$\mathrm{tr}\rho_{\mathcal{A}}(\varepsilon) = e^{-\frac{4\pi^3}{k^2}\frac{\varepsilon}{\ell}\delta_{ij}d^id^j}e^{-\frac{2\pi^3}{3k^2}\frac{\varepsilon}{\ell}}\left(\prod_{j=x,y}\vartheta\left(i4\pi^2\frac{\varepsilon}{\ell}, i\frac{4\pi^2}{k}\frac{\varepsilon}{\ell}d^j\right)\right)\left(\eta\left(i\frac{4\pi^2}{k}\frac{\varepsilon}{\ell}\right)\right)^{-2}, \tag{113}$$

where $\vartheta(\tau,\zeta) = \sum_{z\in\mathbb{Z}}e^{i\pi\tau z^2 + i2\pi\zeta z}$ is the Jacobi theta function and $\eta(\tau)$ is the Dedekind eta function. Recalling the modular properties of these functions, $\vartheta(\tau,\zeta) = (-i\tau)^{-1/2}e^{-\frac{i\pi}{\tau}\zeta^2}\vartheta(-1/\tau,\zeta/\tau)$ and $\eta(\tau) = (-i\tau)^{-1/2}\eta(-1/\tau)$, the leading terms in $\varepsilon/\ell$ are then

$$\mathrm{tr}\rho_{\mathcal{A}}(\varepsilon) = k^{-1}e^{\frac{k}{24\pi}\frac{\ell}{\varepsilon}} + \dots, \tag{114}$$

leading to an entanglement entropy of

$$S_{\mathcal{A}} = \frac{k}{24\pi}\frac{\ell}{\varepsilon} - \log k + \dots. \tag{115}$$

The universal term independent of the cutoff is precisely twice the topological entanglement of Abelian topological order with $k$ anyons. We reproduce this result with an edge mode calculation in the following section.

## 5.2 The edge theory partition function

Because there is no more difficulty in doing so, we will work with a generic "chiral Lifshitz" boundary action appearing in 2.2 and relate the answer to tensor Chern-Simons theory (and the more general $q$-tensor theories) at the end. A typical boundary action from 2.2 is

$$S_z = \frac{k_z}{4\pi}\int dt ds\, \partial_t\partial_s^{\frac{z-1}{2}}\varphi\partial_s^{\frac{z+1}{2}}\varphi, \tag{116}$$

with $z$ odd. The conjugate momentum to $\varphi$ is

$$\pi = \partial_s^z \varphi \,. \tag{117}$$

These fields have engineering dimensions $[\varphi] \sim \ell^{\frac{z-1}{2}}$ and $[\pi] \sim \ell^{-\frac{z+1}{2}}$ such that $k_z$ remains dimensionless. Canonical quantization proceeds by promoting $\varphi$ and $\pi$ to operators with commutation relations

$$[\hat{\pi}(t,s_1), \hat{\varphi}(t,s_2)] = i \frac{2\pi}{k_z} \delta(s_1 - s_2) \,. \tag{118}$$

When considering the bulk entanglement entropy, the boundary action arises on a regulated entangling surface when computing the $n^{th}$ Rényi entropy through a path-integral (see the figure 7 for a cartoon). This entanglement calculation can then be viewed as a path-integral of the edge-mode theory with Euclidean time cycle of length $2\pi n\varepsilon$ where $\varepsilon$ is the regulator of the entangling surface. This is exactly the thermal partition function at inverse temperature $\beta = 2\pi n\varepsilon$ and we are interested in the "high-temperature" ($\varepsilon \to 0$) limit. It is clear to see that (116) has no Hamiltonian and such a path-integral is bound to diverge. This is the same divergence we encountered in our Ishibashi states in section 5. To account for this we need to supplement a Hamiltonian. The lowest order term we can add to the action is

$$H = v_z \frac{k_z}{4\pi} \int ds \, \partial_s^{\frac{z+1}{2}} \varphi \, \partial_s^{\frac{z+1}{2}} \varphi \,, \tag{119}$$

where $v_z$ is a dimensionless coupling. In fact this Hamilonian stems from a *conformal symmetry* generated by the stress tensor

$$T(s) = \frac{k_z}{8\pi} : \partial_s^{\frac{z+1}{2}} \varphi \, \partial_s^{\frac{z+1}{2}} \varphi : \tag{120}$$

obeying the operator product

$$T(s_1)T(s_2) \sim \frac{2T(s_1)}{(s_1 - s_2)^2} + \frac{\partial_s T(s_1)}{(s_1 - s_2)} + \frac{1}{2} \frac{1}{(s_1 - s_2)^4} \,, \tag{121}$$

and $H$ is proportional to the $L_0$ of this stress-tensor. Canonically quantizing, the operator equations of motion are

$$(\partial_t - v_z \partial_s) \partial_s^z \varphi(t,s) = 0 \,. \tag{122}$$

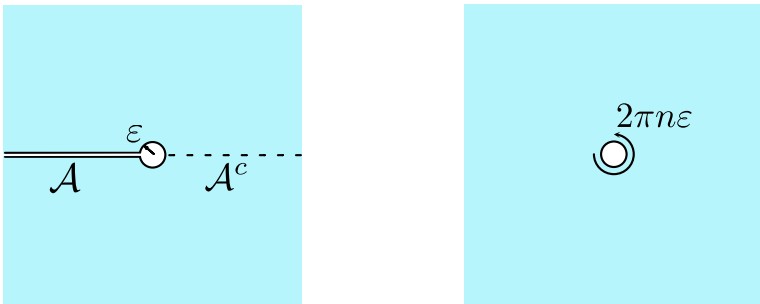

Figure 7: (Left) The Euclidean path-integral preparation of the reduced density matrix on $\mathcal{A}$ is regulated by cutting a $\varepsilon$ keyhole around the entangling surface (which runs out of the page). (Right) The $n^{th}$ Rényi entropy is computed by path-integral with a cylinder of circumference $2\pi n\varepsilon$ excised around the entangling surface. For the $q$-tensor Chern-Simons theory the action entirely pulls back to this cylinder.

Taking the spatial slice to be a circle of circumference $\ell$, the oscillator spectrum of $\varphi$ has frequency $\omega_n = v_z \frac{2\pi n}{\ell}$. There are potentially modes annihilated exactly by $\partial_s^z$ which are not single-valued on the circle. In keeping with $\varphi$'s appearance as a boundary mode from (19), we will require $A_{i_1 \dots i_q}$ or $\partial_s^{\frac{z+1}{2}} \varphi$ to be single-valued on the circle but will allow the winding[19]

$$\partial_s^{\frac{z-1}{2}} \varphi(s+\ell) = \partial_s^{\frac{z-1}{2}} \varphi(s) + 2\pi m, \qquad m \in \mathbb{Z}. \tag{123}$$

The mode expansion of $\varphi$ is then

$$\hat{\varphi}(t,s) = \left\{ \hat{\varphi}_0 + \frac{2\pi}{\ell} \hat{p}_0 \left( t + \frac{2}{z+1} s \right) \right\} \frac{1}{\left( \frac{z-1}{2} \right)!} s^{\frac{z-1}{2}} + \sqrt{\frac{2\pi}{\ell}} \sum_{m=1}^{\infty} \left( \hat{a}_m e^{i\omega_m t + i \frac{2\pi m}{\ell} s} + \hat{a}_m^\dagger e^{-i\omega_m t - i \frac{2\pi m}{\ell} s} \right), \tag{124}$$

with commutators[20]

$$[\hat{\varphi}_0, \hat{p}_0] = i k_z^{-1}, \qquad [\hat{a}_m^\dagger, \hat{a}_n] = k_z^{-1} \left( \frac{2\pi m}{\ell} \right)^{-z} \delta_{m,n}, \tag{125}$$

and from (123) the eigenvalues of $\hat{p}_0$ are integers. The normal-ordered Hamiltonian is then

$$H = \frac{\pi v_z k_z}{\ell} \hat{p}_0^2 + v_z k_z \sum_{m=1}^{\infty} \left( \frac{2\pi m}{\ell} \right)^{z+1} \hat{a}_m^\dagger \hat{a}_m. \tag{126}$$

The partition function, $Z_\beta = \mathrm{tr} e^{-\beta H}$, over the Fock space of the $\hat{a}_m^\dagger$ oscillators is easily evaluated as

$$Z_\beta = \sum_{p_0 \in \mathbb{Z}} e^{-\frac{\beta}{\ell} \pi v_z k_z p_0^2} \prod_{m=1}^{\infty} \sum_{M_m=0}^{\infty} e^{-\frac{\beta}{\ell} 2\pi v_z m M_m} = \vartheta \left( i \frac{\beta}{\ell} v_z k_z \right) \eta^{-1} \left( i \frac{\beta}{\ell} v_z \right) e^{-\frac{\beta}{\ell} \frac{\pi v_z}{12}}. \tag{127}$$

This partition function can be compared to the trace of the reduced density matrix from section 5. At high temperatures $\left( \frac{\beta}{\ell} \to 0 \right)$ the leading terms are

$$Z_\beta = |k_z|^{-1/2} e^{\frac{\pi}{12 v_z} \frac{\ell}{\beta}} + \dots, \tag{128}$$

with an entropy

$$S_\beta = \frac{\pi}{6 v_z} \frac{\ell}{\beta} - \frac{1}{2} \log |k_z|. \tag{129}$$

Recalling that two such scalar theories appear as the edge theory of (19) with couplings $|k_{z_1}| = |k_{z_2}| = k \, 2^{q-2}$ we propose that the entanglement entropy of the $q$-tensor Chern-Simons theory takes the form

$$S^{q\text{-tensor}} = \mathcal{C} \frac{\ell}{\varepsilon} - \log(k \, 2^{q-2}), \tag{130}$$

for some non-universal constant $\mathcal{C}$. When $q = 2$ this matches our result from the extended Hilbert space, (115).

---

[19]If we are interested in making contact with the discussion in section 4, if the group element $g = \exp\left( i \mu^{\frac{z-1}{2}} \varphi \right)$ is compact and $\mu^{-1}$ is a lattice-spacing, "derivatives" on the lattice have compactification radius $(\partial_s)^p \varphi \approx \mu^p (\hat{\Delta}_s)^p \varphi \sim \hat{\Delta}_s \varphi + 2\pi \mu^{p + \frac{1-z}{2}}$.

[20]Actually to be precise about the winding mode, we are reproducing the commutator $[\partial_s^{\frac{z-1}{2}} \varphi, \partial_s^{\frac{z+1}{2}} \varphi] = i \frac{2\pi}{k_z} \delta^2(s_1 - s_2)$.

# 6 Discussion

In this paper we have investigated a class of continuum fracton models characterized by conservation of multipole moments with a focus on the gapped model of dipole conservation in the form of a two-index tensor Chern-Simons theory. We were able to show that gauge-invariant operators in this theory, much like ordinary Chern-Simons theory, must be extended. There are standard string-like operators, but there are also novel strip-like operators whose restricted deformability encodes the underlying fractonic physics. We have also shown that this model possess many similarities to conventional topological order: gapped bulk spectrum, gapped edge spectrum, and "anyon" statistics encoded by the string operators wrapping dipolar defects. An appealing physical picture of this topological order as a dipolar condensate emerges when allowing dipole moments to be quantized by an invariance under a novel set of large gauge transformations. This mechanism requires the introduction of a fundamental length scale which we regard as a sign of the continuum theory remembering the lattice. In this condensate, fractional dipole moments regain mobility and act as the anyon excitations. We bolstered this claim by calculating the entanglement of the ground state and showing that it has the topological correction consistent with two sets of Abelian topological order.

### $U(1)$ vs. $\mathbb{R}$

An essential ingredient to the above discussion is the ability to quantize charges (and in particular dipole moments) in units of some fundamental length scale. We argued in section 4 that this can be seen as a consequence of regarding the gauge symmetry as a compact $U(1)$ as opposed to $\mathbb{R}$. This compact symmetry required treating gauge parameters as lying on an underlying lattice and was motivated from compact global symmetries of related microscopic models. In this sense we can regard the calculations of section 5 as a "lattice/continuum hybrid," extracting the universal corrections to the entanglement of a lattice model equipped with compact global symmetry using continuum gauge theory techniques. There is a certain elegance to this approach as it allows some agnosticism about the specifics of the lattice model[21] and only taking its symmetry constraints as input.

It is natural to also speculate on the alternative: requiring a strictly continuum theory and regarding the gauge symmetry as non-compact. To our awareness, there is no inconsistency with this theory. It also has a gapped bulk spectrum and gapless edge modes. Charges are not necessarily quantized (and in fact, without a lattice scale dipole charge *cannot* be quantized) and the vacuum no longer has the interpretation of a condensate. As such, charge defects are always immobile and dipoles retain the restriction on their mobility to directions orthogonal to their dipole moment. One might call such a phase a *fractonic insulator*. We can also characterize the ground state entanglement in this phase by the same extended Hilbert space method of section 5 and removing the sum over large-gauge identified dipoles. The result is

$$S_{\mathcal{A}}^{(\text{non-compact})} = \frac{k}{24\pi}\frac{\ell}{\varepsilon} - \log\left(\frac{\ell}{\varepsilon}\right) + \text{const.} + \dots. \tag{131}$$

The appearance of a logarithm of the geometric cutoff $\varepsilon$ means that the constant correction is non-universal. The coefficient of the log, $(2 \times \frac{1}{2})$, is universal[22] yet very coarse: it signals that there are two independent bosonic degrees of freedom of the bulk theory. We expect the same

---

[21]Indeed, the task of realizing the tensor Chern-Simons action with a specific lattice Hamiltonian is still unclear. We thank Kevin Slagle for this comment. However, see [44] for the possibility of realizing the mobility constraints of dipole and trace-quadrupole conservation from the hydrodynamics of two-dimensional superfluid vortices.

[22]While this model is fairly different, this coefficient is similar in nature to a universal correction to the entanglement of ordinary (that is, not higher-rank) $U(1)$ spin-liquids [45].

answer for the *q*-tensor theories as well (following the calculation in section 5.2 and removing the sum of windings).

### Topological entanglement for fracton phases

We have seen that the ground state entanglement of this model effectively characterizes the presence of topological order in its ground state, and it does so in a way that mimics the subleading constants of conventional Abelian topological order. This might run counter to general intuitions regarding fracton phases: because symmetries can be associated to subsystems, one might expect features fixed by those symmetries to be extensive. One aspect of this is the relative simplicity of this model as a description of fracton order: as mentioned in the introduction (1), the rotationally symmetric point of the tensor gauge theory breaks subsystem symmetries to a finite set of $U(1)$'s. Another important aspect of our result is the special role of the $\mathbb{R}^2$ background in establishing the analogue to quantum Hall physics: unlike the Hilbert space of ordinary Chern-Simons theory, the Hilbert space of the tensor gauge theory is sensitive to curvature[23] and so our results are particular to the $\mathbb{R}^2$ background. One might imagine that the theory defined on curved space (along the lines of [26]) will display an entanglement entropy sensitive to the geometry. We leave this for future investigation.

The above alludes to a broader question as to what entanglement signatures characterize other fracton orders. There has been notable work in this area in the domain of stabilizer models [46, 47] but it is still largely open area of research. We hope that some of the extended Hilbert space methodology of this paper can be useful for other fracton phases. Perhaps the most obvious suggestion is to investigate the ground state entanglement of *gapless* tensor gauge theories in $(3 + 1)$-d. To our knowledge there is no strict proposal on what the entanglement entropy of such phases looks like much less what universal fractonic/topological corrections to expect. We expect much of our analysis here to be broadly applicable to the "edge mode" contribution to entanglement, however we must also take into account a "bulk" entanglement contribution coming from dynamical tensor-photons (a complication that does not occur in the gapped phase). There are many techniques for evaluating these contributions in Maxwell electrodynamics both canonically [48] and by path-integral [49, 50] and it is feasible for those techniques to work here as well. We regard this as an interesting and tractable direction for the future.

Perhaps more tantalizing is the further investigation of gapped fractonic phases in $(3+1)$-d. Indeed, while the broad Type I/II classification exists, it is suffice to say that we are far from a unified classification or description of these phases. One might hope that entanglement can provide some needed insight. In $(3 + 1)$-d gapped fracton phases some care is needed in distinguishing between the extensive and universal sub-leading corrections to the entanglement entropy [51, 52], however much of the intuition is based on particular stabilizer realizations. It would be very interesting to explore to the character of these subleading corrections directly through an effective field theory calculation of entanglement entropy (using [18] or [19], for example). We plan to address these questions in the near future.

**Acknowledgements:** I would like to thank Onkar Parrikar for many lengthy and enlightening discussions during the duration of this project. I would also like to acknowledge conversations with Aron Wall regarding chiral Lifshitz theories before the onset of this work. Lastly, I would like to extend thanks to Kevin Slagle and Michael Pretko for helpful comments and to Tarek Anous for a careful proofreading of a draft of this paper. This work is supported by the ERC Starting Grant GenGeoHol.

---

[23]We thank A. Gromov for a discussion on this point.

# A   On the universality of $-\frac{1}{2}\log k$

As mentioned in various sections, when the bare theory has no dynamics the sums appearing in edge mode calculations need to be regulated by the addition of a Hamiltonian. When adding the lowest order Hamiltonian consistent with symmetries we arrive at the schematic answer

$$Z_\beta = \vartheta(k\tau)\eta^{-1}(\tau)e^{i\frac{\pi}{12}\tau}, \qquad \tau \sim i\frac{\varepsilon}{\ell}, \tag{A.1}$$

for each independent bosonic field pulled back to the entangling surface (in this case the independent chiral Lifshitz modes or the independent dipole moments). The appearance of a $-\frac{1}{2}\log k$ in the entropy comes from a partial cancellation of $(-ik\tau)^{-1/2}$ in the high-temperature expansion of $\vartheta$ (stemming from the winding sum) with the $(-i\tau)^{-1/2}$ in the high-temperature expansion of the $\eta$ (stemming from the oscillator sum). In this appendix we will address whether additional higher-derivative interactions in the Hamiltonian can alter this behavior at high-temperatures. We claim that the answer is no: that while additional interactions can affect area law terms, the $-\frac{1}{2}\log k$ is uniquely determined by the lowest-order Hamiltonian.

Let us suppose to (119) we add higher-derivative interactions:

$$H^{\text{high der}} = \frac{v_z k_z}{4\pi}\int ds : \partial_s^{\frac{z+1}{2}}\varphi\,\partial_s^{\frac{z+1}{2}}\varphi : + \sum_{p\geq 1}\frac{\gamma_p}{2M^{2p}}\int ds : \partial_s^{\frac{z+1}{2}+p}\varphi\,\partial_s^{\frac{z+1}{2}+p}\varphi :, \tag{A.2}$$

for a some mass-scale $M$. This modifies the chiral Lifshitz partition function, (127), to

$$Z_\beta^{\text{high der}} = \text{tr}\,e^{-\beta H^{\text{high der}}} = \vartheta\left(\frac{ik_z\sigma}{2\pi}\right)\prod_{m=1}^{\infty}\sum_{N_m=0}^{\infty}e^{-\sigma(m+\sum_{p=1}\alpha_p m^{2p+1})N_m}$$

$$= \vartheta(k_z\tau)\prod_{m=1}^{\infty}\left(1-e^{-\sigma\left(m+\sum_{p=1}\alpha_p m^{2p+1}\right)}\right)^{-1}, \tag{A.3}$$

where $\sigma = \frac{2\pi\beta v_z}{\ell}$ and $\alpha_p = \frac{(2\pi)^{2p+1}\gamma_p}{k_z v_z(M\ell)^{2p}}$ (these details will not be important to the main result). Let us define

$$\Phi_f(\sigma) := -\log\prod_{m=1}^{\infty}\left(1-e^{-\sigma f_m}\right), \qquad f_m = m + \sum_{p=1}\alpha_p m^{2p+1}. \tag{A.4}$$

The $-\frac{1}{2}\log k$ correction to the entropy will be universal if there is no logarithmic dependence on the cutoff in $\log Z_\beta$. This requires the $\sigma\to 0$ limit of $\Phi_f(\sigma)$ to behave as

$$\lim_{\sigma\to 0}\Phi_f(\sigma) = \frac{1}{2}\log\left(\frac{\sigma}{2\pi}\right) + \text{Laurent polynomial in }\sigma, \tag{A.5}$$

to cancel a similar log appearing from the theta function. Expanding the log in (A.4)

$$\Phi_f(\sigma) = \sum_{m=1}^{\infty}\sum_{n=1}^{\infty}\frac{1}{n}e^{-n\sigma f_m}, \tag{A.6}$$

and using the inverse Mellin transform of $e^{-n\sigma f_m}$

$$\Phi_f(\sigma) = \sum_{m=1}^{\infty}\sum_{n=1}^{\infty}\frac{1}{2\pi in}\int_{c-i\infty}^{c+i\infty}du\,\Gamma(u)(\sigma n f_m)^{-u}$$

$$= \frac{1}{2\pi i}\int_{c-i\infty}^{c+i\infty}du\,\Gamma(u)\zeta(u+1)\zeta_f(u)\sigma^{-u}, \tag{A.7}$$

where

$$\zeta_f(u) = \sum_{m=1}^{\infty} f_m^{-u} = \sum_{m=1}^{\infty} \left( m + \sum_p \alpha_p m^{2p+1} \right)^{-u}, \tag{A.8}$$

is the zeta function associated to $f_m$ and $c \in \mathbb{R}_+$ is large enough so that the sums converge uniformly (i.e. to the right of the pole of $\zeta_f(u)$ with largest real part). Note that the unperturbed zeta function (with all $\alpha_p = 0$) is the Riemann zeta, $\zeta(u)$. If we are interested in the leading terms in the $\sigma \to 0$ limit we can follow the method of [53]: we imagine pushing the $u$ contour to $-1 < c < 0$ such that the resulting integrand vanishes in the $\sigma \to 0$ limit. In doing so however, we will pass through the double pole of $\Gamma(u)\zeta(u+1)$ at $u = 0$ and the poles of $\zeta_f(u)$. We expect $\zeta_f(u)$ to at least have a simple pole[24] at $u = \frac{1}{2p_{max}+1}$ where $p_{max}$ is the maximum higher derivative term added to the action. This residue of this pole modifies the polynomial behavior of $\sigma$ in the $\sigma \to 0$ limit. The $\log \sigma$ (and constant terms) affecting the bulk topological entanglement however comes from the residue of poles at $u = 0$. If $\zeta_f(u)$ does not have a pole at $u = 0$ then the residue there gives

$$\text{Res}_{u \to 0} \Phi_f(\sigma) = \zeta_f'(0) - \zeta_f(0) \log(\sigma). \tag{A.9}$$

Thus the topological correction is universal if $\zeta_f(u)$ is regular at $u = 0$ and $\zeta_f(0) = \zeta(0) = -\frac{1}{2}$ and $\zeta_f'(0) = \zeta'(0) = -\frac{1}{2}\log 2\pi$. Let us argue this is true by a formal perturbative expansion[25] of (A.8):

$$\zeta_f(u) - \zeta(u) = \sum_{m=1}^{\infty} m^{-u} \sum_{n=1}^{\infty} \frac{\Gamma(1-u)}{\Gamma(n+1)\Gamma(1-n-u)} \left( \sum_p \alpha_p m^{2p} \right)^n. \tag{A.10}$$

A typical term of the above is

$$\zeta_f(u) - \zeta(u) \supset \frac{\Gamma(1-u)}{\Gamma(n+1)\Gamma(1-n-u)} \zeta(u - N_n), \tag{A.11}$$

where $N_n$ is an even integer depending on $n$. Investigating the behavior close to $u \approx 0$, we find that this term has a double zero for each $n \geq 1$: one from $\frac{1}{\Gamma(1-n)}$ and another from $\zeta(-N_n)$ since $N_n$ is even. Thus both $\zeta_f(u) - \zeta(u)$ and $\zeta_f'(u) - \zeta'(u)$ vanish on each term of the perturbation theory.

## B   Details on large gauge transformations

In this appendix we provide some details on the large gauge transformations discussed in section 4. We do these calculations carefully because $\theta$ is not a linear variable and derivatives generally do not commute when acting on it: $\varepsilon^{ij}\partial_i\partial_j\theta = 2\pi\delta^2(\vec{x}-\vec{x}_o)$. Because of this multiple derivatives acting on even the vanilla large gauge transformation $\alpha_0 = \mu^{-1}\theta$ can result in delta function contributions. This is well illustrated by the variation of the "flux" under $\alpha_0$:

$$\varepsilon^{ij}\partial_i\partial^k \left( \delta_{\alpha_0} A_{jk} \right) = \frac{\mu^{-1}}{2} \varepsilon^{ij} \left( \partial_i\partial^k\partial_j\partial_k\theta + \partial_i\partial^k\partial_k\partial_j\theta - \partial_i\partial_j\partial^2\theta \right)$$
$$= \frac{\mu^{-1}}{2} \varepsilon^{ij}\partial^2\partial_i\partial_j\theta$$
$$= \mu^{-1}\pi\partial^2\delta^2(\vec{x}-\vec{x}_o), \tag{B.1}$$

---

[24] That is the large $m$ behavior of $\zeta_f$ is $\alpha_{p_{max}}^{-u} m^{-(2p_{max}+1)u} \sim \alpha_{p_{max}}^{-u} \zeta((2p_{max}+1)u)$ which has the ordinary pole of the Riemann zeta function at $(2p_{max}+1)u = 1$.

[25] This expansion obviously does not converge generically. This is perhaps to be expected and there is physics in this statement. Indeed, if we imagine approximating $\zeta_f(s)$ by performing the $m$ sum up to a large maximum $m_{max}$ then the perturbation theory fails if for one of the couplings, $\alpha_p m_{max}^{2p} \sim 1$. This is the usual energy scale where perturbative effective field theory breaks down: $\omega_{max} = \frac{2\pi m_{max}}{\ell} \sim M$.

where in the first line we used explicitly the symmetric combination of derivatives (as dictated by $A_{jk}$ being a symmetry tensor) and going from the first to the second line we used that derivatives commute when acting on the divergence of $\theta$ (i.e. $\partial_i \theta = -\varepsilon_{ij} \frac{(x-x_o)^j}{(x-x_o)^2}$ is a covector with periodic components) which kills the first and third terms. The two derivatives appearing in (B.1) indicate that the charge and the dipole moment of a region containing $\vec{x}_o$, $\Sigma$, are invariant under $\alpha_0$, but the trace-quadropole moment is not:

$$\delta_{\alpha_0} \int_{\Sigma} d^2x \, x^2 \, \varepsilon^{ij} \partial_i \partial^k A_{jk} = 4\pi \mu^{-1} \,, \tag{B.2}$$

which accounts for the phase acquired by $\mathbf{T}_\nu$, (82). For the other two of the three large gauge transformations from section 4 to investigate:

$$\alpha_0 = \mu^{-1}\theta \,, \qquad \alpha_1^i = \theta \, x^i \,, \qquad \alpha_2 = \mu\theta x^2 \,, \tag{B.3}$$

we will find it easier to look at the action of $\alpha_1^i$ and $\alpha_2$ on string operators directly, as opposed to their action on the flux. This is because derivatives of $\theta$, once pulled back to the contour of a string operator, $\mathcal{C}_s$, commute as long as $\mathcal{C}_s$ does not contact the origin of the $\theta$ coordinate. With this in mind, we will need the ingredients $\delta_\alpha A_{ij}$, $\delta_\alpha \partial^j A_{ij}$, and $\delta_\alpha \varepsilon^{ij} \partial_i A_{jk}$ showing up in the exponents of the string operators for $\alpha = \alpha_1^i$, $\alpha_2$. We will use the notation $\widehat{=}$ to indicate "equal when pulled back to $\mathcal{C}_s$." For $\alpha_1^p$ we find

$$\delta_{\alpha_1^p} A_{ij} \widehat{=} \partial_i \partial_j \theta \, x^p + \partial_i \theta \delta_j^p + \partial_j \theta \delta_i^p - \delta_{ij} \partial^p \theta \,,$$
$$\partial^j (\delta_{\alpha_1^p} A_{ij}) \widehat{=} \partial^p \partial_i \theta \,,$$
$$\varepsilon^{ki} \partial_k (\delta_{\alpha_1^p} A_{ij}) \widehat{=} -\varepsilon^k{}_j \partial_k \partial^p \theta \,, \tag{B.4}$$

where we've noted $\partial^2 \theta \widehat{=} 0$. For $\alpha_2$ we have

$$\delta_{\alpha_2} A_{ij} \widehat{=} \mu \left( \partial_i \partial_j \theta \, x^2 + 2\partial_i \theta x_j + 2\partial_j \theta x_i - 2\delta_{ij} x^k \partial_k \theta \right) \,,$$
$$\partial^j (\delta_{\alpha_2} A_{ij}) \widehat{=} \mu \left( 2x^j \partial_i \partial_j \theta + 4\partial_i \theta \right) \,,$$
$$\varepsilon^{ki} \partial_k (\delta_{\alpha_2} A_{ij}) \widehat{=} -4\mu\varepsilon^k{}_j \partial_k \theta - 2\mu\varepsilon^k{}_j x^\ell \partial_k \partial_\ell \theta \,. \tag{B.5}$$

Plugging these expressions into (57) we find

$$\delta_{\alpha_1^p} \log \mathbf{M}_p = 0 \,,$$
$$\delta_{\alpha_2} \log \mathbf{M}_p = i\frac{p}{2} \oint_{\mathcal{C}_s} ds^i \left( 2x^j \partial_i \partial_j \theta + 4\partial_i \theta \right) = i2\pi p \,, \tag{B.6}$$

and into (59)

$$\delta_{\alpha_1^p} \log \mathbf{D}_{\bar{\mathbf{v}}} = i\mathbf{v}^j \oint_{\mathcal{C}_s} ds^i \left( \partial_i \partial_j \theta \bar{x}^p + \partial_i \theta \delta_j^p + \partial_j \theta \delta_i^p - \delta_{ij} \partial^p \theta \right) + i\mathbf{v}_\ell \oint_{\mathcal{C}_s} ds \, \hat{n}^j \bar{x}^\ell \varepsilon^k{}_j \partial_k \partial^p \theta$$
$$= i2\pi \mathbf{v}^p \,,$$
$$\delta_{\alpha_2} \log \mathbf{D}_{\bar{\mathbf{v}}} = i\mu\mathbf{v}^j \oint_{\mathcal{C}_s} ds^i \left( \partial_i \partial_j \theta \, \bar{x}^2 + 2\partial_i \theta \, \bar{x}_j + 2\partial_j \theta \, \bar{x}_i - 2\bar{x}^k \partial_k \theta \, \delta_{ij} \right)$$
$$+ i\mu\mathbf{v}_\ell \oint_{\mathcal{C}_s} ds \, \hat{n}^j \bar{x}^\ell \left( 4\varepsilon^k{}_j \partial_k \theta + 2\varepsilon^k{}_j \bar{x}^m \partial_k \partial_m \theta \right)$$
$$= 0 \,. \tag{B.7}$$

In the above we used $ds\,\hat{n}^j\,\varepsilon^k{}_j\,\partial_k(\cdot) = -ds^k\partial_k(\cdot)$ based on $\hat{n}$'s definition as the outward normal to $\mathcal{C}_s$. Lastly, plugging into (62)

$$
\begin{aligned}
\delta_{\alpha_1^p}\log\mathbf{T}_\nu &= i\mu\nu\oint_{\mathcal{C}_s} ds^i\,\bar{x}^j\left(\partial_i\partial_j\theta\,\bar{x}^p + \partial_i\theta\delta_j^p + \partial_j\theta\delta_i^p - \delta_{ij}\partial^p\theta\right) + \frac{i}{2}\mu\nu\oint_{\mathcal{C}_s} ds\,\bar{x}^2\hat{n}^j\varepsilon^k{}_j\partial_k\partial^p\theta \\
&= 0\,, \\
\delta_{\alpha_2}\log\mathbf{T}_\nu &= i\mu^2\nu\oint_{\mathcal{C}_s} ds^i\bar{x}^j\left(\partial_i\partial_j\theta\bar{x}^2 + 2\partial_i\theta\bar{x}_j + 2\partial_j\theta\bar{x}_i - 2\delta_{ij}\bar{x}^k\partial_k\theta\right) \\
&\quad + \frac{i\mu^2\nu}{2}\oint_{\mathcal{C}_s} ds\hat{n}^j\bar{x}^2\left(4\varepsilon^k{}_j\partial_k\theta + 2\varepsilon^k{}_j\bar{x}^m\partial_k\partial_m\theta\right) \\
&= 0\,.
\end{aligned}
\tag{B.8}
$$

We summarize these results in the following table.

|  |  | $\alpha_0$ | $\alpha_1^p$ | $\alpha_2$ |
|---|---|---|---|---|
| $\delta\log\mathbf{M}_p$ |  | 0 | 0 | $i2\pi p$ |
| $\delta\log\mathbf{D}_{\bar{\mathbf{v}}}$ |  | 0 | $i2\pi\mathbf{v}^p$ | 0 |
| $\delta\log\mathbf{T}_\nu$ |  | $-i2\pi\nu$ | 0 | 0 |

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
