# Peer review of "Entanglement in the quantum Hall fluid of dipoles"

_SciPost Physics, doi:SciPost Phys. 11, 052 (2021)_

## Round 1 · Referee Report · Anonymous (Referee 1) · 2021-7-21

Report

The present manuscript is a technical work, where rank-2 Chern-Simons theory is quantized on a plane and then the entanglement entropy is calculated using a method that goes back to the work of Buividovich and Polikarpov. In this method the Hilbert space of a gauge theory is enlarged to include non-gauge invariant states, but has simple, tensor product structure.

The first four sections of the work, is revision and repackaging of mostly known results and serves as a nice introduction to a reader not familiar with the field. Particularly nice is discussion of the large gauge transformations.

The main new result is presented in section 5. I did not attempt to reproduce the calculation of the entanglement entropy however, I expect that it is correct. What gives me confidence is the relation of the answer to the known result for the EE of the U(1)_k Chern-Simons theory. On a plane rank-2 theory can be written as U(1)_k Chern-Simons theory with two gauge fields. This was pointed out in Ref. [1] of the manuscript. Thus the naïve answer for the EE would be twice the EE of U(1)_k Chern-Simons theory, which appears to be the case and was shown rigorously.
The sub-leading correction is the well-known topological entanglement entropy and is determined by the quantum dimension D, S = logD. In the present case D = k^{1/2} * k^{1/2} which give S = log(k). Consequently I believe it is correct as well, and also is shown rigorously.

Finally, I would like to highlight the calculation of the EE using the edge theory, which in the present case is non-relativistic Lifshitz theory. I think this calculation is important and, perhaps, should be moved to the main text. The reason is the following. While formally one might guess (in view of analogy between two copies of U(1)_k CS theory and rank-2 CS theory) that the edge theory is two chiral bosons, in reality the edge theory is one ordinary scalar boson and one non-relativistic chiral boson. Still, the two theories give the same EE, which is not obvious a priori.

---

## Round 1 · Referee Report · Anonymous (Referee 2) · 2021-8-4

Strengths

The topics are interesting. Most of the calculations are presented in detail. The author covered most of the questions of quantizing the CS theory of traceless symmetric tensor gauge on R^3.

Weaknesses

There are still some technical steps that are missed.

Report

This manuscript proposed a path integral quantization procedure for the Chern-Simon action of a version of the higher-rank gauge theory, the traceless symmetric tensor gauge. The author also studied in detail the topological defects and proposed the quantization condition for multipole charges.

The topics of this manuscript are intriguing and timely, providing that the higher-rank symmetry attracts a vast of attention from both condensed matter and high energy communities. I would recommend this manuscript published on SciPost Physics if the author could clarify the following issues/questions satisfactorily.

1) There is no explicit definition of N(δ[]s) in the manuscript.
2) When changing the path-integral variable from Aij to ϕ (From Eq (36) to Eq (43)), do we need to worry about the explicit form of the Jacobian, or the Jacobian was automatically taken care of?

3) I can see the examples strip operators in Figs 2(a)-2(d) satisfy the conditions (71) and (73). Could the author demonstrate explicitly that examples 2(e) and 2(f) also meet those conditions?

4) Extra: The manuscript proposed a quantization procedure for CS theory of a higher-rank gauge on a general space Σ, but only the results for ΣR2 are derived explicitly. Can one extend the calculations to non-trivial topological space (Torus T2, for example)?

Other comments:
A) The restoration of mobility of fractonic excitations was studied previously in the context of fracton/elasticity duality in SciPost Phys. 9, 076 (2020), Phys. Rev. B 100, 045119 (2019), Phys. Rev. Lett. 121, 235301 (2018). In the dual picture, the condensation of dipoles (dislocations) corresponds to quantum melting. One can choose to condense dislocations (dipoles) in one direction to achieve the smectic phase or condense dislocations (dipoles) in both direction to obtain the nematic phase.
B) The direct connection between quantum Hall physics and the traceless symmetric tensor gauge theory was proposed in arXiv:2103.09826. One can use the higher-rank gauge symmetry to analyze the fractonic behaviours of low energy excitations in Fractional Quantum Hall systems.

Minor:
There could be a typo in equation (43)

---

## Round 2 · Referee Report · Anonymous (Referee 2) · 2021-8-8

Report

The author replied that there is no typo in equation (43), then I don't understand the meaning of $q^a(\partial \bar{\partial})^2(\mathbf{z}_a ,\mathbf{z}_b)q^b$.

Regarding examples in Figs (2e) and (2f), I have my last question to the author. Could the cases happen in a lattice? Do they violate the quantization of dipole moment? Of course, my question doesn't affect the claims of the manuscript since the author begins discussing the quantization in section 4.

Other than that I am satisfied with the author's answers. I recommend the paper to be published on Scipost Physics.
  • validity: good
  • significance: good
  • originality: good
  • clarity: high
  • formatting: excellent
  • grammar: excellent

Author:  Jackson Fliss  on 2021-08-08  [id 1645]

(in reply to Report 1 on 2021-08-08)
Category:
answer to question

Thank you to the anonymous report for the insightful questions. With regards to these questions:

1) On equation (43): $(\partial\bar{\partial})^{-2}$ is a distribution (the inverse of $(\partial\bar{\partial})^2$) which takes two spatial points as its argument. These are evaluated at the locations of the defects (${\bf z}_a$) because the source, $\rho$, appearing in the first line of equation (43) is localized to the point of the defects.

2) On figures (2e) and (2f): I do not think these strip operators can, strictly speaking, be realized on a lattice although some approximation to them is possible depending on the details of the lattice. If so then I do think their lattice approximation will not violate dipole quantization.

---

## Round 2 · Author Response

I thank the referees for their helpful comments and I have implemented their recommendations in this resubmission. I have also double checked the calculation up to and including equation (43) mentioned by referee 1 and have verified that there is no typo.

---

## Round 2 · List of Changes

1) added a sentence (below equation (40)) defining $\mathcal N (\delta[]'s)$ 2) added a footnote (footnote 7) explaining that the Jacobian from changing path-integral variables A_{ij}->\phi is canceled by an inverse determinant from the delta function constraint. 3) Added explicit parameterizations to figure 2(a)-(f) and modified caption of figure 2 so that it is clear that the examples satisfy the conditions for gauge invariance. 4) Expanded the first paragraph of section 2.1 to clarify to the reader for when results hold for R^2 and when they hold on general surface. Added sentence to footnote 6 to explain how quantization on T^2 differs and added relevant reference. 5) Updated first paragraph of section 4.1 to mention the prior work on crystal melting as suggested by referee 2. 6) Updated second paragraph of page 4 to mention prior work noting quantum Hall physics and tensor gauge theory as suggested by referee 2. 7)Moved Appendix A (previous draft) to main text as section 5.2 as suggested by referee 1. I have given the bulk gauge theory calculation its own heading ("5.1 The extended Hilbert space") to better distinguish the two approaches.

---

## Editorial Decision

published